# Localization, proteomics, and metabolite profiling reveal a putative vesicular transporter for UDP-glucose

Cheng Qian[1], Zhaofa Wu[2,3†], Rongbo Sun[2,3†], Huasheng Yu[2,3,4], Jianzhi Zeng[2,3,4], Yi Rao[2,3,4,5], Yulong Li[2,3,4,5]*

[1]School of Life Sciences, Tsinghua University, Beijing, China; [2]State Key Laboratory of Membrane Biology, Peking University School of Life Sciences, Beijing, China; [3]PKU-IDG/McGovern Institute for Brain Research, Beijing, China; [4]Peking-Tsinghua Center for Life Sciences, Beijing, China; [5]Chinese Institute for Brain Research, Beijing, China

*For correspondence:
yulongli@pku.edu.cn

†These authors contributed equally to this work

Competing interests: The authors declare that no competing interests exist.

**Abstract** Vesicular neurotransmitter transporters (VNTs) mediate the selective uptake and enrichment of small-molecule neurotransmitters into synaptic vesicles (SVs) and are therefore a major determinant of the synaptic output of specific neurons. To identify novel VNTs expressed on SVs (thus identifying new neurotransmitters and/or neuromodulators), we conducted localization profiling of 361 solute carrier (SLC) transporters tagging with a fluorescent protein in neurons, which revealed 40 possible candidates through comparison with a known SV marker. We parallelly performed proteomics analysis of immunoisolated SVs and identified seven transporters in overlap. Ultrastructural analysis further supported that one of the transporters, SLC35D3, localized to SVs. Finally, by combining metabolite profiling with a radiolabeled substrate transport assay, we identified UDP-glucose as the principal substrate for SLC35D3. These results provide new insights into the functional role of SLC transporters in neurotransmission and improve our understanding of the molecular diversity of chemical transmitters.

## Introduction

The release of extracellular signaling molecules by secretory vesicles is a strategy used by a wide range of cell types and tissues and plays an essential role under both physiological and pathological conditions (*Burgoyne and Morgan, 2003*). A key step in the process is the accumulation of the respective signaling molecules into the secretory vesicles by specific transporter proteins. In the nervous system, vesicular neurotransmitter transporters (VNTs) such as VGLUT and VGAT (which transport glutamate and GABA, respectively) are essential for the transport of small-molecule neurotransmitters into synaptic vesicles (SVs). These selective transporters determine the category, amount, and transport kinetics of neurotransmitters, thereby establishing the molecular basis of the underlying chemical neurotransmission (*Blakely and Edwards, 2012*). All VNTs identified to date belong to the Solute Carrier (SLC) superfamily of membrane transport proteins, the second-largest group of membrane proteins in the human proteome, with more than 400 members spanning 65 subfamilies (http://slc.bioparadigms.org/) (*Hediger et al., 2013*). Strikingly, approximately 30% of these 400 transporters are either uncharacterized or orphan transporters (*César-Razquin et al., 2015*; *Perland and Fredriksson, 2017*), providing the opportunity to identify novel VNTs and their cognate substrates, thus identifying new neurotransmitters and/or neuromodulators.

The physiological role of transporter proteins is closely coupled with their subcellular localization; however, to date, localization profiling of transporters – particularly SLC transporters, including which are expressed on secretory organelles in primary cells – has not been systematically studied.

Tagging a protein of interest with a fluorescent protein is a widely used strategy for localization profiling (*Chong et al., 2015*; *Huh et al., 2003*; *Simpson et al., 2000*), and this approach offers an effective strategy for screening large numbers of targeted proteins. In addition, the development of mass spectrometry (MS)–based proteomics coupled with subcellular fractionation has made it possible to examine the subcellular spatial distribution of the proteome both rapidly and efficiently (*Andersen et al., 2003*; *Christoforou et al., 2016*; *Itzhak et al., 2016*; *Orre et al., 2019*), including the SV proteome (*Laßek et al., 2015*; *Takamori et al., 2006*). Immunoisolation of SVs, followed by proteomic analysis using high-sensitivity MS, provides a specific and efficient method for characterizing the molecular anatomy of SVs (*Boyken et al., 2013*; *Grønborg et al., 2010*) including endogenous SLC transporters.

Electron microscopy (EM) is the gold standard to obtain ultrastructural information since it offers the vastly superior resolution (on the order of 1 nm in biological samples) compared to the resolution of optical imaging (on the order of 200–300 nm) (*Fernández-Suárez and Ting, 2008*). Moreover, using a genetically encoded tag for EM overcomes certain limitations associated with classic immuno-EM labeling methods, which require specific antibodies and penetration of those antibodies. APEX2, an enhanced variant of ascorbate peroxidase, is a highly efficient proximity-based EM tag (*Lam et al., 2015*) suitable for determining the subcellular localization of proteins of interest.

Identifying the molecular function of an orphan transporter is an essential step toward understanding its biological function. However, using the classic radiolabeled substrate transport assay to deorphanize transporters is a relatively low-throughput approach, particularly given the virtually unlimited number of chemicals that can be tested. On the other hand, metabolite profiling using MS is a high-throughput method for knowing the content metabolites (*Chantranupong et al., 2020*; *Nguyen et al., 2014*; *Vu et al., 2017*) that can offer insights into candidate substrates. Thus, combining metabolite profiling together with the radiolabeled substrate transport assay will likely yield new insights into the molecular function of orphan transporters.

The nucleotide sugar uridine diphosphate glucose (UDP-glucose) plays an essential role in glycosylation in both the endoplasmic reticulum and the Golgi apparatus (*Moremen et al., 2012*). Interestingly, the release of UDP-glucose into the extracellular space was detected previously using an enzyme-based method (*Lazarowski et al., 2003*). Subsequent experiments with 1231N1 cells (an astrocytoma cell line) showed that the release of UDP-glucose requires both $Ca^{2+}$ signaling and the secretory pathway, as the release was inhibited by the $Ca^{2+}$ chelator BAPTA and the Golgi apparatus blocker brefeldin A (*Kreda et al., 2008*).

Nucleotide sugars are transported into subcellular organelles by the SLC35 family, which contains 31 members, including 20 orphan transporters (*Caffaro and Hirschberg, 2006*; *Ishida and Kawakita, 2004*; *Song, 2013*). Importantly, the level of nucleotide sugars released by cells can be manipulated by changing the expression of SLC35 transporters; for example, knocking out an SLC35 homolog in yeast decreased the release of UDP-*N*-acetyl-galactosamine, whereas overexpressing human SLC35D2 in airway epithelial cells increased UDP-*N*-acetyl-galactosamine release (*Sesma et al., 2009*). However, whether UDP-glucose is transported by a SLC35 transporter located on secretory organelles is currently unknown.

In this study, we screened 361 SLC members using localization profiling and identified 40 candidate vesicular transporters. In parallel, we performed proteomics analyses of immunoisolated SVs from mouse brain samples and found that seven transporters overlapped, including the orphan SLC35 subfamily transporters SLC35D3, SLC35F1, and SLC35G2. Further ultrastructural analysis using APEX2-based EM supported that the SLC35D3 is capable of trafficking to SVs. Finally, we combined metabolite analysis and the radiolabeled substrate transport assay in subcellular organelles and identified UDP-glucose as the principal substrate of SLC35D3.

## Results

### Identification of a subset of SLC35 proteins as putative vesicular transporters using localization screening of SLC transporters

To identify new candidate vesicular transporters, we performed localization screening of SLC transporters (*Figure 1*). First, we created a cloning library containing 361 human SLC family members fused in-frame with the red fluorescent protein mCherry; we then systematically co-expressed

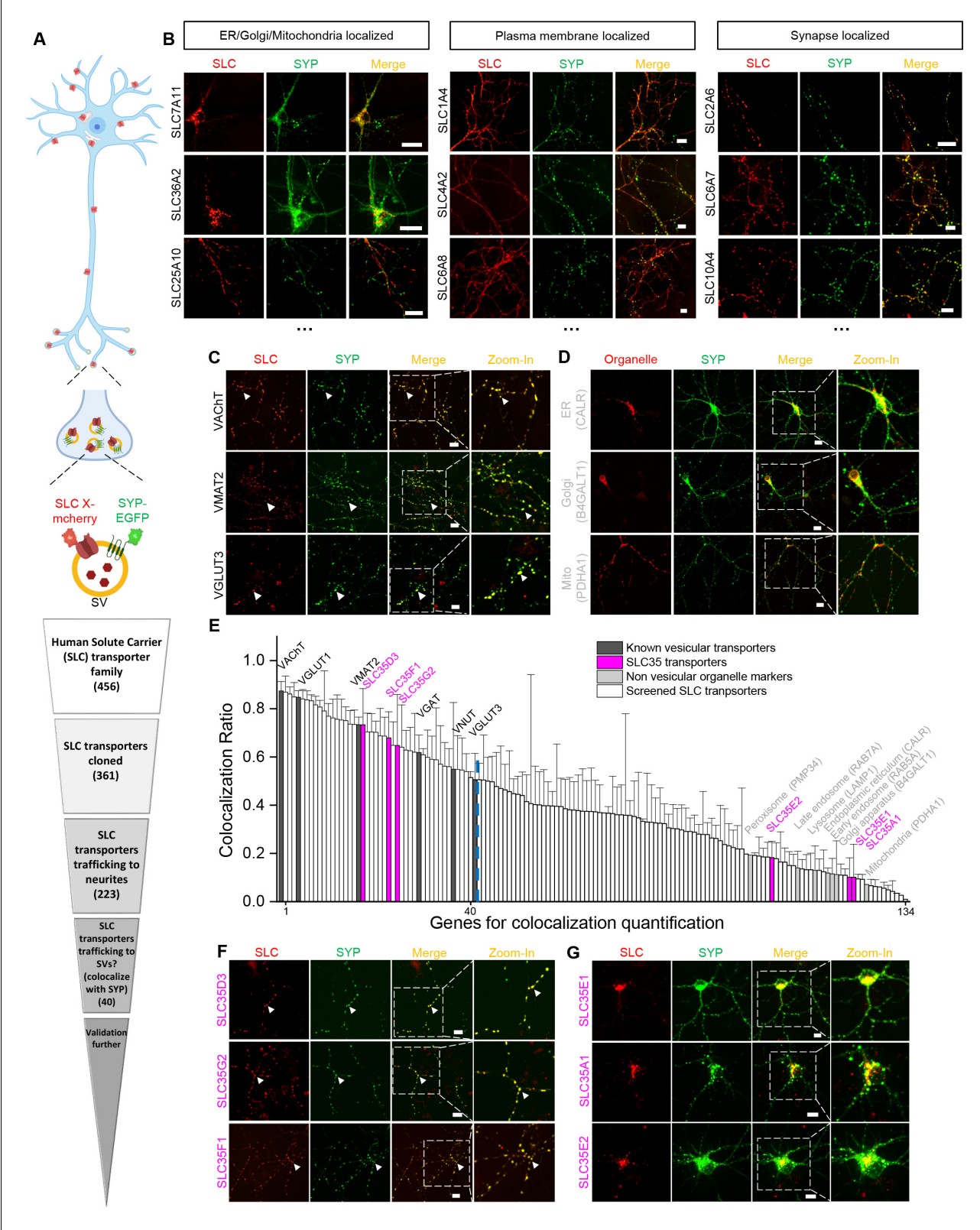

**Figure 1.** Localization profiling of SLC family members reveals candidate vesicular transporters. (**A**) Top: Schematic diagram of the localization profiling strategy. Red and green fluorescent signals were collected using confocal microscopy imaging of cultured rat neurons co-expressing mCherry-tagged SLC proteins and EGFP-tagged synaptophysin (SYP-EGFP). Bottom: Sequential steps used for the localization profiling. Two rounds of screening revealed a total of 40 of 361 screened SLC transporters as candidate vesicular transporters. (**B**) Representative images of neurons expressing SLC

*Figure 1 continued on next page*

*Figure 1 continued*

X-mCherry transporters (red) and SYP-EGFP (green). Scale bars: 10 μm. (**C**) Representative images of neurons expressing three known vesicular SLC transporters (red) and SYP-EGFP (green), with magnified views. White arrowheads indicate co-localization. Scale bars: 10 μm. (**D**) Representative images of neurons expressing three non-vesicular organelle markers (red) and SYP-EGFP (green), with magnified views. Scale bars: 10 μm. (**E**) Summary of the co-localization ratio between 134 proteins and SYP-EGFP. Dark gray bars represent known vesicular transporters, magenta bars represent SLC35 transporters, light gray bars represent non-vesicular organelle markers, and white bars represent the SLC transporters screened in this study. The threshold indicated by the vertical dashed line was defined as the co-localization ratio between VGLUT3 and SYP-EGFP. n = at least three neurons each. Data are mean ± s.e.m. (**F, G**) Representative images of neurons expressing vesicular (**F**) and non-vesicular (**G**) SLC35 transporters (red) and SYP-EGFP (green), with magnified views. White arrowheads indicate co-localization. Scale bars: 10 μm.

The online version of this article includes the following source data and figure supplement(s) for figure 1:

**Source data 1.** Genes for co-localization quantification.

**Figure supplement 1.** Subcellular localization of SLC35D3 at a reduced expression level.

individual SLC-mCherry construct with EGFP-tagged synaptophysin (SYP-EGFP) to label SVs in cultured rat cortical and hippocampal neurons, revealing the localization of SLC transporters (*Figure 1A,B*). Of the 223 SLC transporters that trafficked to neurites, 134 showed overlap with SYP-EGFP and were analyzed further by quantifying the co-localization ratio between the red and green fluorescent signals (*Figure 1A,E*). As expected, known synaptic vesicular transporters such as VGLUT and the vesicular acetylcholine transporter (VAChT) had relatively high co-localization ratio with SYP-EGFP (50–80% co-localization) (*Figure 1C,E*), whereas markers of non-vesicular organelles such as the Golgi apparatus, endoplasmic reticulum, and mitochondria had relatively low co-localization ratio (10–20%) (*Figure 1D,E*). Setting a threshold at the co-localization ratio for VGLUT3 – a well-known vesicular transporter – revealed a total of 40 candidate vesicular transporters (*Figure 1E* and *Supplementary file 1*). Among these candidates, a subset of SLC35 transporters, including SLC35D3, SLC35F1, and SLC35G2, had a co-localization ratio of approximately 70% with SYP-EGFP (*Figure 1E,F*). In contrast, other members of the same subfamily, such as SLC35A1, SLC35E1, and SLC35E2, localized primarily to organelles in the soma and had relatively low co-localization ratio (10–20%) (*Figure 1E,G*). Together, these results indicate that putative vesicular transporters, including a subset of SLC35 family members, likely localize to neuronal SVs.

To avoid mis-localization caused by overexpression, we tested different delivery strategies for a low expression level on one candidate SLC35D3. The lowest expression level of epitope-tagged SLC35D3 was achieved using lentivirus, which was ~40% compared with plasmid transfection (*Figure 1—figure supplement 1A,B*). Then we focused on the localization of SLC35D3 in the lentivirus infected neurons (*Figure 1—figure supplement 1C*). The co-localization ratio between SLC35D3 and SYP (SV marker) was ~60%, which is similar to that in the plasmid transfected neurons (~70%). Given SYP may also be localized to secretory granules, we co-immunostained a secretory granule marker Chg A and found that the co-localization ratio between SLC35D3 and Chg A was ~30%. Taken together, SLC35D3 with relatively low expression level has a higher possibility to localize to synaptic vesicles than to secretory granules.

## Proteomics analysis of SVs reveals novel vesicular transporters

To probe the proteome including the vesicular transporters presented in SVs, we immunoisolated intact SVs from fractionated mouse brain samples and used western blot analysis and high-performance liquid chromatography (HPLC)–MS to analyze the proteome (*Figure 2A*). Using a specific antibody against SYP to isolate SVs, we found a number of SV markers present in the anti-SYP samples, but not in samples obtained using a control IgG (*Figure 2B*); as an additional control, the post-synaptic marker PSD-95 was not detected in either the anti-SYP sample or the control IgG sample in western blotting. Moreover, using EM, we directly observed SVs on the surface of anti-SYP beads, but not control IgG beads (*Figure 2C*), confirming that the anti-SYP beads selectively isolate SVs.

Next, we performed HPLC–MS analysis and found high reproducibility among repeated trials in both the anti-SYP and control IgG samples (*Figure 2—figure supplement 1*). We further analyzed the relatively abundant proteins (LFQ intensity > $2^{20}$, without immunoglobin) that were significantly enriched in the anti-SYP sample compared to the control sample (*Figure 2D*). The proteins enriched in the anti-SYP sample covered more than 60% of the 110 proteins in the SV proteome listed in the SynGO database (*Koopmans et al., 2019*), including known VNTs, vesicular ATPase subunits, and a

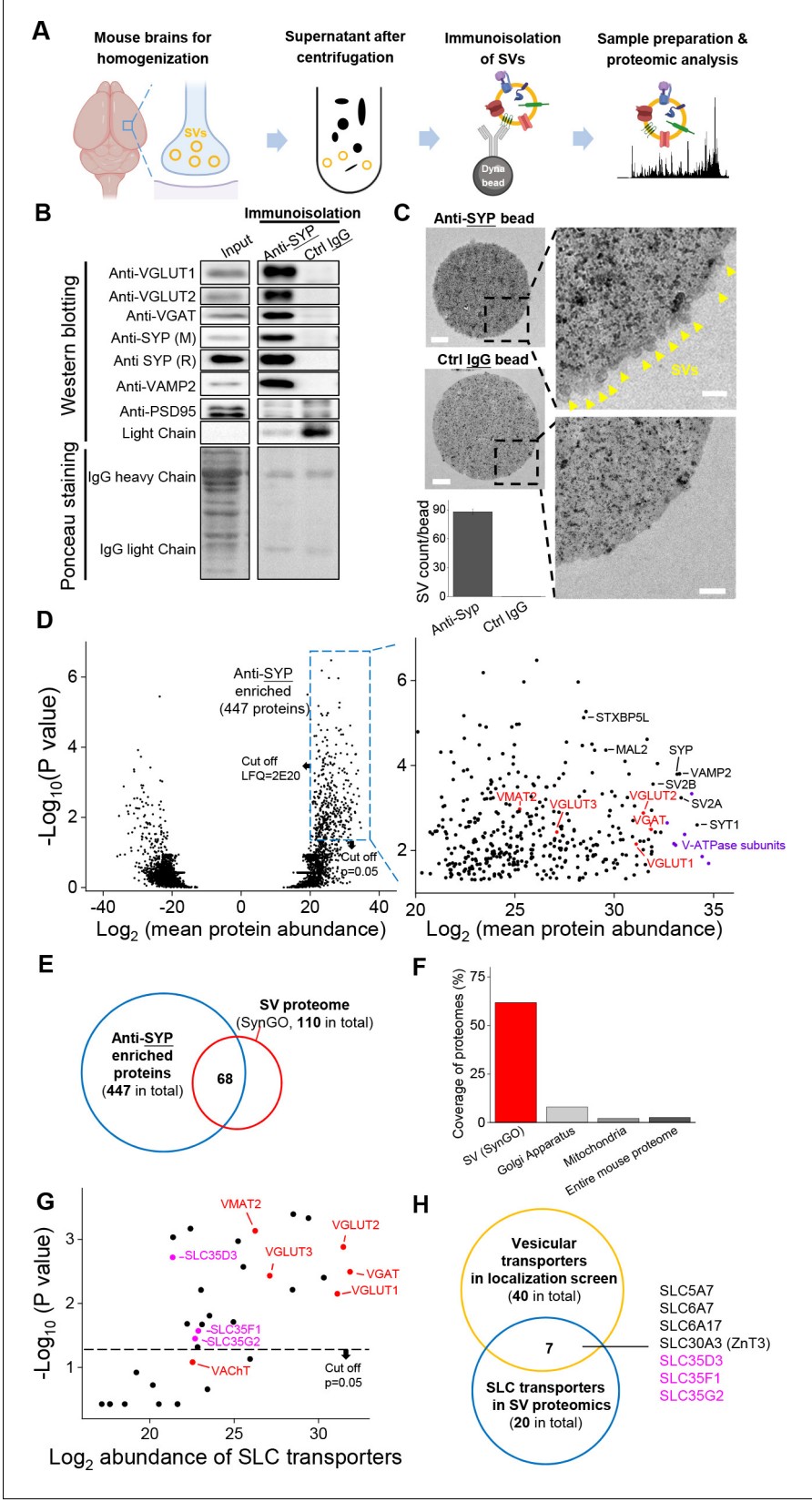

**Figure 2.** Proteomics profiling of SVs identifies novel putative vesicular SLC transporters. (**A**) Schematic diagram depicting the strategy for proteomics profiling of SVs immunoisolated from fractionated mouse brain

*Figure 2 continued on next page*

*Figure 2 continued*

homogenates. (**B**) Top: western blot analysis of the indicated protein markers for SVs and the postsynaptic marker PSD-95 in the input fraction (supernatant after centrifugation of whole brain lysates), the anti-SYP immunoisolated sample, and the control IgG sample. Bottom: Ponceau staining of the membrane, showing the total proteins. (**C**) Electron microscopy images of anti-SYP beads (top) and control IgG beads (bottom), with magnified views. Arrowheads indicate immunoisolated SVs. Scale bars: 500 nm and 100 nm (magnified views). The bottom-left panel shows the quantification of the number of SVs attached to the indicated beads. (**D**) Left: volcano plot depicting the proteins detected using SV proteomics. The blue dashed box indicates anti-SYP-enriched proteins using thresholds set at p<0.05 and LFQ intensity > $2^{20}$. n = 3 independently prepared protein samples. p-values by two-sided Student's t-test. Right: magnified view of the anti-SYP–enriched proteins. Representative SV markers are shown in black, V-ATPase subunits are shown in purple, and known vesicular transporters are shown in red. (**E**) Venn diagram showing the overlap between anti-SYP–enriched proteins (blue) and the known SV proteome based on the SynGO database (red). (**F**) Summary of the percentage of overlap between anti-SYP–enriched proteins and the SV proteome (from the SynGO database), Golgi apparatus proteins (from UniProt), mitochondrial proteins (from UniProt), and the entire mouse proteome (from UniProt). (**G**) Summary of the SLC transporters identified using SV proteomics. Classic VNTs are shown in red, and SLC35 transporters are shown in magenta. p-values by two-sided Student's t-test. The horizontal dashed line indicates the threshold at p=0.05. (**H**) Venn diagram showing the overlap between the vesicular transporters identified using localization profiling (yellow) and the vesicular transporters identified using proteomics profiling of SVs (blue). The three candidate SLC35 transporters are shown in magenta.

The online version of this article includes the following source data and figure supplement(s) for figure 2:

**Source data 1.** Proteomics profiling of SVs.

**Figure supplement 1.** Repeatability of the proteomic data Scatterplots showing the correlation between independent biological trials.

**Figure supplement 2.** SLC35D3 in subcellular fractionation of SVs.

---

number of other SV markers (*Figure 2D–F*). Conversely, only 8.0% and 2.2% of the proteins in the mitochondrial and Golgi apparatus proteome, respectively, were present in the anti-SYP sample (*Figure 2F*), indicating minimal contamination by these organelles; as an additional control, we found very little overlap between the proteins in the anti-SYP sample and the entire mouse proteome in the UniProt database (*UniProt Consortium et al., 2019*).

We then focused on SLC transporters and identified 20 SLC transporters, including SLC35D3, SLC35F1, and SLC35G2, among the SV-associated proteins (*Supplementary file 2*). The abundance of these three transporters was similar to known VNTs, including VAChT and the monoamine transporter VMAT2 (*Figure 2G*), even though VAChT was below the threshold for significance (p>0.05). Comparing the putative vesicular transporters identified in our localization screen with the SLC transporters identified in the SV proteome revealed a total of seven transporters present in both datasets, including the three SLC35 family members (i.e., SLC35D3, SLC35F1, and SLC35G2) identified above (*Figure 2H*). The other four transporters were previously reported to localize to SVs including the choline transporter SLC5A7 (*Ferguson et al., 2003*; *Nakata et al., 2004*; *Ribeiro et al., 2003*), the proline transporter SLC6A7 (*Crump et al., 1999*; *Renick et al., 1999*), the neutral amino acid transporter SLC6A17 (*Fischer et al., 1999*; *Masson et al., 1999*), and the zinc transporter SLC30A3 (*Wenzel et al., 1997*).

To further dissect the subcellular distribution of one novel vesicular transporter candidate, SLC35D3, in different organelles, we performed differential centrifugation to fractionate the mouse brain (*Huttner et al., 1983*; *Figure 2—figure supplement 2A*). Firstly, we conducted retro-orbital injection of AAV-PhP.eB virus to infect the mouse brain (*Challis et al., 2019*). The expression of epitope-tagged SLC35D3 was detected 3 weeks after AAV injection (*Figure 2—figure supplement 2B*). With the progress of differential centrifugation, we observed enrichment of SLC35D3 from P2' (crude synaptosome) to LP2 (crude SVs) fraction, which is similar to known SV markers VGLUT1 and SYP. In contrast, the secretory granule marker Chg A, organelle markers of ER and Golgi are majorly enriched before P2' (*Figure 2—figure supplement 2C*). SLC35D3 and VGLUT1 also appeared in P1 and S1 fractions, likely due to the reason that these membrane proteins are being produced and processed through the secretory pathway. In summary, these data corroborate the view that SLC35D3 is less likely to be a classic ER/Golgi transporter and tends to localize to SVs.

## Localization of SLC35D3 to SVs revealed by EM

To further verify the vesicular localization of one of the three SLC35 candidates, SLC35D3, we used APEX2-based labeling (*Lam et al., 2015*) coupled with EM (*Figure 3A*). We first validated this strategy by transfecting cultured rat neurons with Mito-APEX2 to label mitochondria and found two distinct populations based on electron density (*Figure 3B*); as an additional control, we found only one population of SVs in non-transfected neurons (*Figure 3C*). Importantly, neurons transfected with either VGLUT1-APEX2 (*Figure 3D*) or SLC35D3-APEX2 (*Figure 3E*) had two distinct populations of SVs based on electron density, demonstrating that SLC35D3 could localize to SVs in cultured neurons.

## Deorphanization of SLC35D3 using metabolite profiling combined with a radiolabeled substrate transport assay

To search for the cognate substrate corresponding to the orphan vesicular transporter SLC35D3, we used metabolite profiling, based on the assumption that overexpressing the transporter will enrich its cognate substrate in organelles. In our analysis, we intentionally focused on nucleotide sugars present in mammals as possible substrates, as the SLC35 transporter family has been reported to transport these molecules (*Figure 4A*; *Caffaro and Hirschberg, 2006*; *Ishida and Kawakita, 2004*; *Song, 2013*). By optimizing a hyperPGC column–based HPLC strategy coupled with selected reaction monitoring in MS (*Garcia et al., 2013*), we successfully detected a range of nucleotide sugars (*Figure 4B*). Next, we used the deorphanization strategy shown in *Figure 4C*. Firstly, we measured nucleotide sugars in untransfected control cells, finding all known nucleotide-sugars (*Figure 4D,E*). To test the sensitivity of this deorphanization strategy, we generated a stable cell line overexpressing EGFP-tagged SLC35A2 (*Figure 4—figure supplement 1A*), which is known to transport the nucleotide sugars including UDP-galactose and UDP-*N*-acetyl-galactosamine (*Ishida et al., 1996*; *Segawa et al., 2002*; *Sun-Wada et al., 1998*). Profiling the relative abundance of specific nucleotide sugars in organelles prepared from control cells and SLC35A2-overexpressing (SLC35A2OE) cells revealed a >100% increase in the substrate UDP-galactose in SLC35A2OE organelles (*Figure 4F,G*, *Figure 4—figure supplement 1B*). Interestingly, we also detected 60% higher levels of UDP-glucose in SLC35A2OE cells, indicating a previously unknown substrate of the SLC35A2 transporter; in contrast, we found that the SLC35A2 substrate UDP-*N*-acetyl-galactosamine did not appear to be enriched in SLC35A2OE cells, possibly due to limitations in separating UDP-*N*-acetyl-glucosamine and UDP-*N*-acetyl-galactosamine in our HPLC–MS setup (*Figure 4F,G*). We then used this same strategy to search for substrates of the orphan vesicular transporter SLC35D3 using SLC35D3-overexpressing (SLC35D3OE) cells (*Figure 4—figure supplement 1A*). Our analysis revealed a 40% increase in UDP-glucose and a 30% increase in CMP-sialic acid in SLC35D3OE organelles compared to control organelles (*Figure 4H,I*, *Figure 4—figure supplement 1B*), suggesting that these two nucleotide sugars might be substrates of the SLC35D3 transporter.

Metabolite profiling can detect the effects of both direct transport activity and indirect changes in the abundance of metabolites due to the overexpression of transporters; thus, we also conducted an uptake assay with radiolabeled nucleotide sugars in order to measure the transport activity (*Figure 5A*). We found that cells expressing the SLC35A2 transporter had significantly increased uptake of both the previously known substrate UDP-galactose and the newly identified substrate UDP-glucose compared to control cells (*Figure 5B*), validating our deorphanization strategy combining metabolite profiling and the radiolabeled transport assay. Importantly, cells expressing human SLC35D3 had a nearly onefold increase in UDP-glucose transport, but no significant change in the transport of UDP-galactose or UDP-N-acetyl-glucosamine; similar results were obtained from the cells expressed the mouse SLC35D3 (*Figure 5B*). Thus, UDP-glucose is a promising substrate of SLC35D3.

## Characterization of the transport properties of SLC35D3

Next, we characterized the transport of UDP-glucose by SLC35D3. To study the substrate specificity of SLC35D3, we performed a competition assay in which we applied a 100-fold higher concentration of non-radiolabeled substrate together with radiolabeled UDP-glucose in the transport assay. We found that non-radiolabeled UDP-glucose – but not the structurally similar UDP-*N*-acetyl-galactosamine – virtually eliminated the transport of radiolabeled UDP-glucose (*Figure 5C*). In addition,

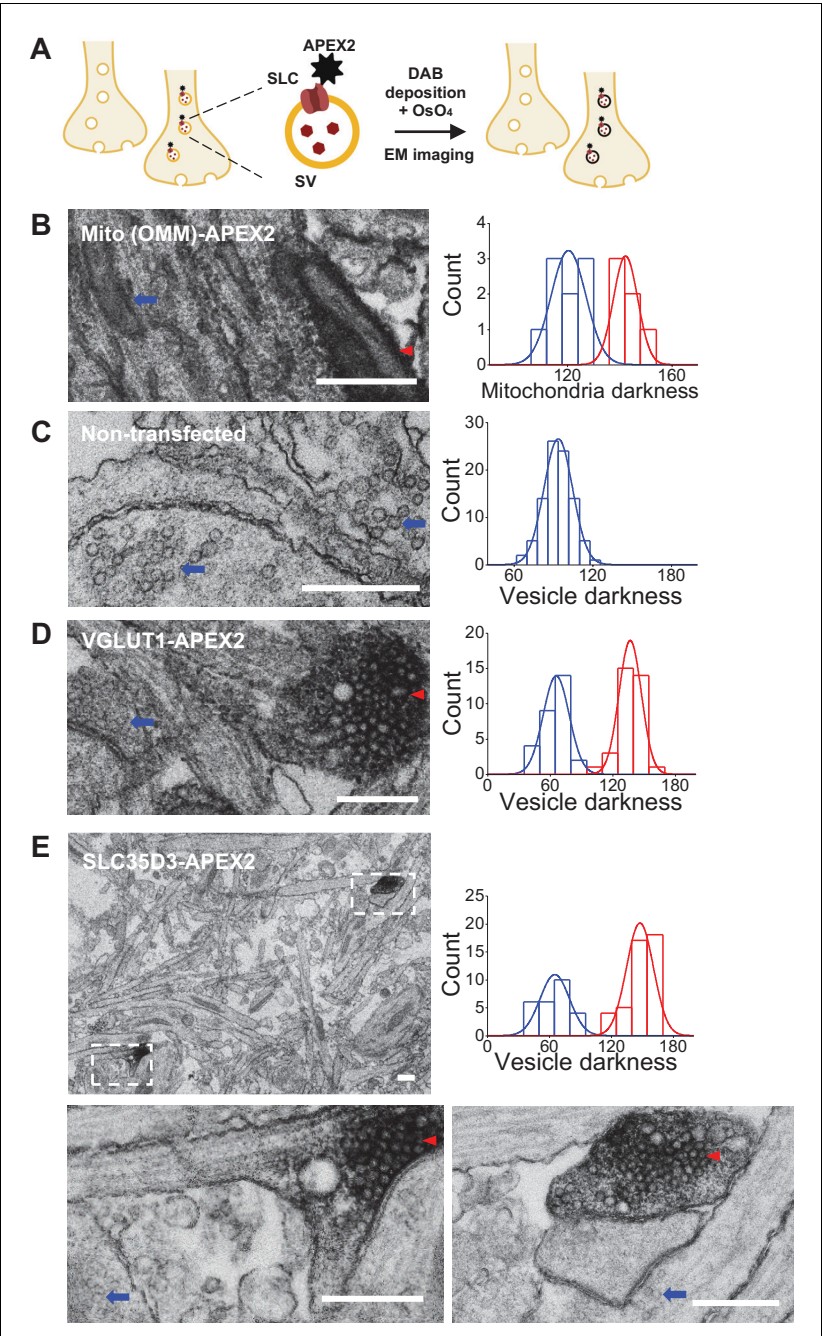

**Figure 3.** Validation of the vesicular localization of SLC35D3 using electron microscopy. (A) Schematic diagram depicting the APEX2-based labeling strategy for studying ultrastructural localization. (B–E) Representative EM images (left) and distribution of organelle darkness (right) of mitochondria in cultured rat neurons transfected with Mito-APEX2 (B), SVs in non-transfected neurons (C), and SVs in neurons transfected with either VGLUT1-APEX2 (D) or SLC35D3-APEX2 (E), with magnified views of the dashed boxes from (E). The blue arrows and red arrowheads indicate organelles with low (light) and high (dark) electron density, respectively. Scale bars: 500 nm.
The online version of this article includes the following source data for figure 3:

**Source data 1.** Quantification of organelle darkness.

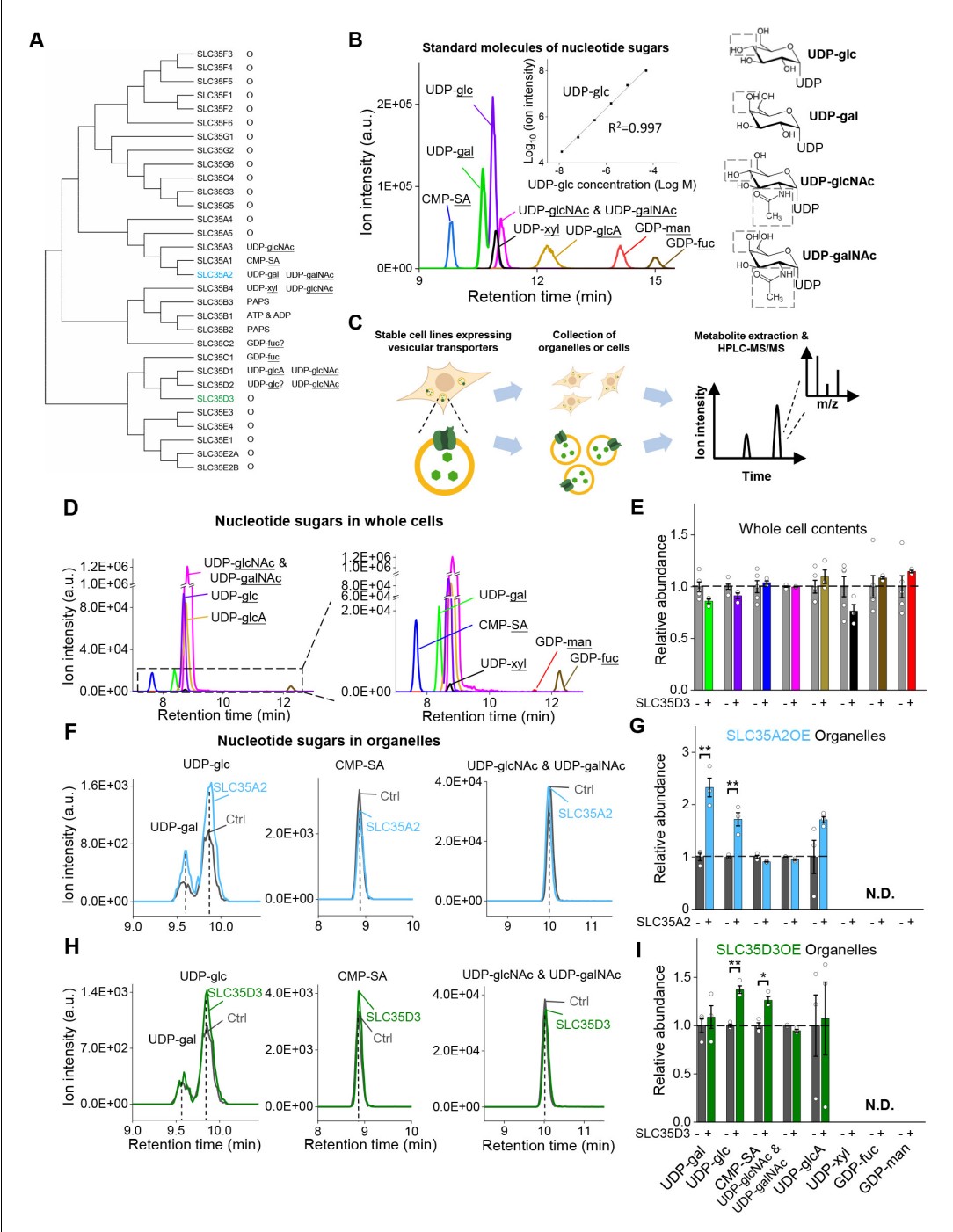

**Figure 4.** The targeted metabolite profiling reveals putative substrates of SLC35D3. (**A**) Phylogenic tree of the SLC35 transporter family and known corresponding substrates. SLC35A2 and SLC35D3 are shown in blue and green, respectively. O: orphan transporters. (**B**) Left: representative HPLC-MS trace showing 5 μM of the indicated nucleotide sugars. The inset shows the linear correlation between the UDP-glc standard and MS ion intensity ($R^2 = 0.997$, Pearson's r). Right: molecular structures of the UDP-sugars UDP-glc, UDP-gal, UDP-glcNAc, and UDP-galNAc, with differences shown in the gray dashed boxes. (**C**) Schematic diagram depicting the strategy for detecting metabolites in organelles and in whole cells. (**D**) Representative traces of the indicated nucleotide sugars detected in control (SLC35A2KO) cells, with a magnified view at the right. (**E**) Summary of the relative abundance of the indicated nucleotide sugars measured in control cells and cells overexpressing SLC35D3. n = 5 and 3 independently prepared metabolite extracts, respectively. (**F–G**) Representative extracted ion chromatograms of specific nucleotide sugars (**F**) and summary of their relative abundance (**G**) in organelles isolated from control cells (gray) and cells overexpressing SLC35A2 (blue). n = 3 independently prepared metabolite extracts. p-values by two-sided Student's t-test. p=0.0049 for UDP-gal and p=0.0099 for UDP-glc abundance, respectively. N.D.: not detectable. (**H–I**) Representative extracted ion chromatograms of specific nucleotide sugars (**F**) and summary of their relative abundance (**G**) in organelles isolated from control cells

*Figure 4 continued on next page*

*Figure 4 continued*

(gray) and cells overexpressing SLC35D3 (green). n = three independently prepared metabolite extracts. p-values by two-sided Student's t-test. p=0.00196 for UDP-glc and p=0.01006 for CMP-SA abundance, respectively. N.D.: not detectable. Data are mean ± s.e.m.; two-sided Student's t-test.

The online version of this article includes the following source data and figure supplement(s) for figure 4:

**Source data 1.** Targeted metabolite profiling of nucleotide sugars.
**Figure supplement 1.** Additional analysis of metabolite profiling.

several other UDP-sugars partially inhibited transport activity but were not enriched in the metabolite profiling, possibly by competing with UDP-glucose on the transporter's substrate-binding pocket. Interestingly, CMP-sialic acid did not reduce the transport of UDP-glucose (*Figure 5C*), even though this nucleotide sugar was increased – albeit to a lesser extent than UDP-glucose – in the organelles of cells expressing SLC35D3 (see *Figure 4I*), indicating that CMP-sialic acid may not be a direct substrate of SLC35D3 but may have been indirectly increased on its abundance as shown by metabolite profiling.

We also measured the time course and dose dependence of UDP-glucose transport by SLC35D3, revealing a time constant of 2.9 min (*Figure 5D*) and a Km value of 0.87 µM (*Figure 5E*). Lastly, we examined the role of the electrochemical proton gradient on SLC35D3 activity, as this gradient has been reported to drive the activity of known VNTs (*Edwards, 2007*; *Van Liefferinge et al., 2013*). We therefore applied a variety of pharmacological inhibitors and measured UDP-glucose transport by SLC35D3 (*Figure 5F*). We found that *N*-ethylmaleimide (NEM), FCCP (carbonyl cyanide-4-(trifluoromethoxy) phenylhydrazone), and nigericin significantly reduced UDP-glucose transport in SLC35D3-expressing cells (*Figure 5G*), suggesting that the electrochemical proton gradient contributes – at least in part – to the driving force. Interestingly, Bafilomycin A1 did not reduce the transport. Unlike the proton uncouplers that directly abolish the proton electrochemical gradient, Bafilomycin A1 inhibits V-ATPase that indirectly affects the maintenance of proton electrochemical gradient (*Yoshimori et al., 1991*). There can be preserved proton electrochemical gradient in SVs after the acute application of Bafilomycin A1, as indicated by a previous work using pH-dependent quantum dots to study SV Kiss and Run (K and R) and full-collapse fusion (FCF) (*Zhang et al., 2009*), which may support UDP-glucose transport by SLC35D3.

To compare the transport mechanism of SLC35D3 with a canonical ER/Golgi localized SLC35 transporter, we investigated the pharmacological properties of SLC35A3, which is an ER/Golgi localized UDP-N-acetyl-glucosamine transporter (*Ishida et al., 1999*; *Figure 5—figure supplement 1A*). We found the pharmacological treatment including proton uncouplers did not significantly inhibit UDP-N-acetyl-glucosamine transport, indicating SLC35A3 may have a different transport mechanism compared with SLC35D3 (*Figure 5—figure supplement 1B*). Moreover, the transport activity of GDP-mannose by a yeast homolog of the nucleotide–sugar transporters was neither sensitive to CCCP nor valinomycin (*Parker and Newstead, 2017*), which also suggested different transport mechanisms among nucleotide–sugar transporters. Further studies by proteoliposome reconstitution of purified SLC35D3 can help to illustrate the detailed transport mechanism, e.g., if SLC35D3 has the obligate exchanger activity. Taken together, these data support the notion that SLC35D3 is a nucleotide sugar transporter, with UDP-glucose as its primary substrate.

## Discussion

Here, we report the identification and characterization of three novel SLC35 transporters putatively localized to SVs using a combination of localization profiling, proteomics profiling, and EM. Using metabolite profiling combined with a radiolabeled substrate transport assay, we also found that one of these novel vesicular transporters – SLC35D3 – is a UDP-glucose transporter. These data indicate the potential existence of a novel neuronal vesicular transporter of the nucleotide sugar UDP-glucose (*Figure 6*).

Our localization screening strategy revealed a series of vesicular transporter candidates in neurons, a cell type that has tightly regulated secretory vesicles. We cannot rule out the possibility that these transporters may also play a physiological role in regulated secretory granules in non-neuronal secretory cells. Taking the well-known vesicular transporter VMAT2 as an example, it could localize

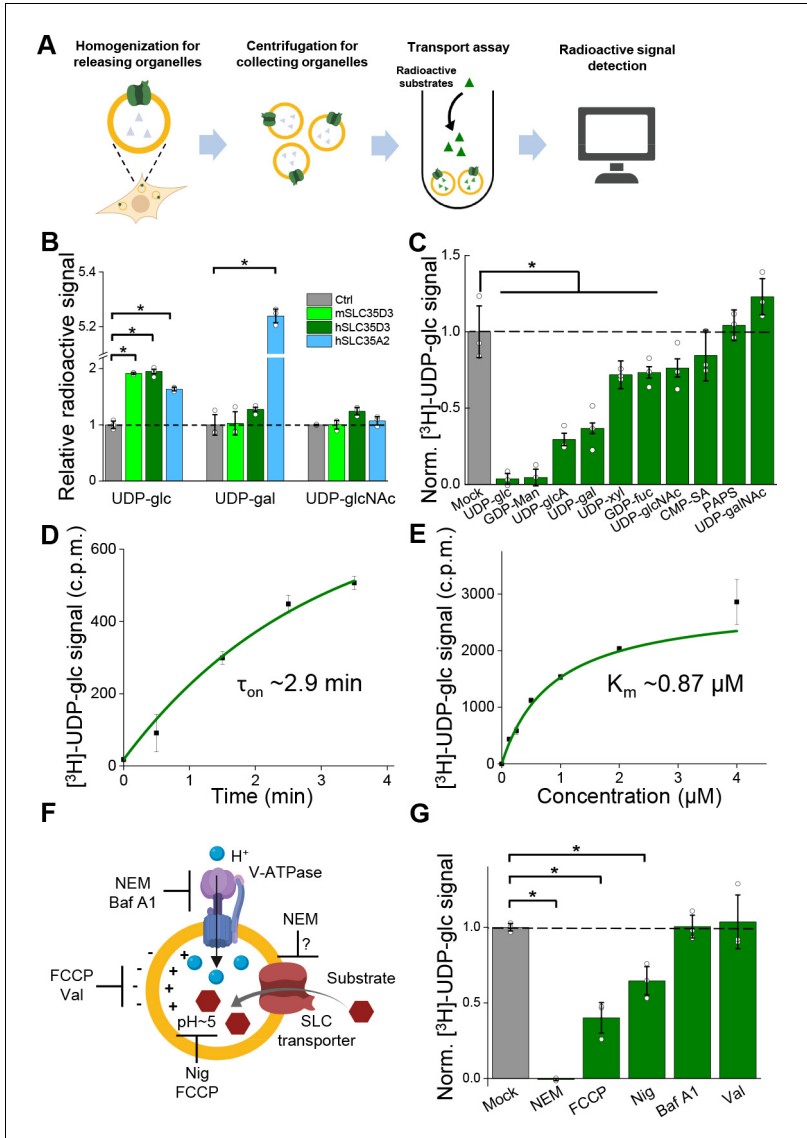

**Figure 5.** Validation and characterization of the UDP-glucose transport activity of SLC35D3. (**A**) Schematic diagram depicting the transport assay using organelles isolated from HEK293T cells. (**B**) Summary of the transport of [3H]-UDP-glc, [3H]-UDP-gal, and [3H]-UDP-glcNAc (500 nM each) in control (SLC35A2KO) cells and in cells overexpressing mouse SLC35D3 (mSLC35D3), human SLC35D3 (hSLC35D3), or human SLC35A2 (hSLC35A2); n = 3 experiments each. p=0.04953 for mSLC35D3, hSLC35D3, and hSLC35A2 in UDP-glc transport; p=0.04953 for hSLC35A2 in UDP-gal transport. (**C**) Competition assay measuring [3H]-UDP-glc (500 nM) transport in the presence of the indicated non-labeled compounds (at 50 μM) in cells expressing SLC35D3; the data are expressed relative to mock cells, in which solvent was applied instead of a non-labeled compound; n = 3 experiments each. p=0.04953 for cold UDP-glc, GDP-Man, UDP-glcA, UDP-gal, UDP-xyl, and GDP-fuc competition. (**D**) Time course of [3H]-UDP-glc transport measured in cells expressing SLC35D3, relative to corresponding baseline values. The data were fitted to a single-exponential function. (**E**) Dose–response curve for [3H]-UDP-glc transport in cells expressing SLC35D3, relative to the corresponding baseline values. The data were fitted to Michaelis–Menten kinetics equation. (**F**) Schematic diagram depicting the proton gradient driving vesicular transporters, with specific inhibitors shown. (**G**) Summary of [3H]-UDP-glc transport measured in cells expressing SLC35D3, expressed relative to mock cells, in which solvent was applied; n = 3 experiments each. NEM, *N*-ethylmaleimide (0.2 mM); FCCP, carbonyl cyanide-4-(trifluoromethoxy) phenylhydrazone (50 μM); Nig, Nigericin (5 μM); Baf, bafilomycin A1 (100 nM); Val, valinomycin (20 μM). p=0.04953 for NEM, FCCP, and Nig inhibition. Data are mean ± s.e.m.; p-values by Kruskal–Wallis ANOVA test.

The online version of this article includes the following source data and figure supplement(s) for figure 5:

*Figure 5 continued on next page*

*Figure 5 continued*

**Source data 1.** UDP-glucose transport activity of SLC35D3.
**Figure supplement 1.** Pharmacology-insensitive transport activity by SLC35A3.

to both synaptic vesicles and large dense-core vesicles in neurons (*Nirenberg et al., 1996*), as well as secretory granules in endocrine cells of the peripheral system (*Weihe et al., 1994*).

It is important to note that some VNTs may have been below the detection limit of enriched proteins in our SV proteomics approach. For example, the vesicular nucleotide transporter SLC17A9 has been reported to play a role in vesicular ATP release (*Sawada et al., 2008*), but was not identified in our proteomics analyses of SVs, consistent with reports by other groups (*Grønborg et al., 2010*; *Takamori et al., 2006*). Similarly, our analysis did not identify SLC10A4, another vesicular transporter (*Larhammar et al., 2015*). Therefore, studies regarding these low-abundance transporters may require more robust strategies such as enriching specific SVs from VNT-expressing brain regions, using specific antibodies against VNTs, or generating transgenic mice expressing biochemical labels on specific VNTs.

In addition to our study, a subset of SLC35 family members was also reported by SV proteomics. SLC35G2 was recently reported in SV proteomics using an improved workflow (*Taoufiq et al., 2020*). Interestingly, SLC35D3 was not simultaneously identified, potentially due to a few reasons: (1) the proteome may vary across different species at different ages (SD rats at 4–6 weeks vs C57BL6 mice at 6–8 weeks); (2) SLC35D3 has an even lower protein abundance compared with SLC35G2 (*Figure 2G*), which is more challenging for detection; and (3) different purification strategies may lead to differences in SV pools. For example, another SLC35 family member, SLC35F5, was found to be enriched in VGAT-positive SVs instead of VGLUT1-positive SVs, even though the majority of the two proteomes were highly similar (*Boyken et al., 2013*; *Grønborg et al., 2010*). Taken together, these studies provided hints for identifying vesicular SLC35 transporters.

Biochemical fractionation strategies (e.g., differential fractionation and density gradient fractionation) combined with antibodies recognizing endogenous proteins are classic in validating the subcellular localization of the protein of interest. Given limited performance of antibodies in detecting SLC35D3, we tried exogenous delivery of SLC35D3 using AAV-PhP.eB, which infected the whole mouse brain efficiently therefore providing satisfactory starting materials. It is worth noting that AAV-PhP.eB potentially results in overexpression of SLC35D3 in the brain that may affect the subcellular distribution of the transporter. In addition, the LP2 fraction (crude SVs) after differential fractionation may contain other organelles such as secretory granules and lysosomes. Subsequent studies using more efficient SLC35D3 antibodies and further purified SVs can be of help to validate the localization of endogenous SLC35D3 in vivo.

Combining metabolite profiling with a radiolabeled substrate transport assay is a powerful tool for identifying and characterizing transporter substrates (*Nguyen et al., 2014*; *Vu et al., 2017*), which could facilitate the classic deorphanization of an orphan transporter by screening the costly and environmentally unfriendly radioactive ligands in transport assay. Therefore, targeted candidates in metabolic profiling were in a higher priority for further validation like radioactive transport assay. Here, we show that this strategy can indeed be effective for studying orphan vesicular transporters located on secretory organelles. The performance of metabolic profiling and the transport assay is largely dependent on the signal-to-

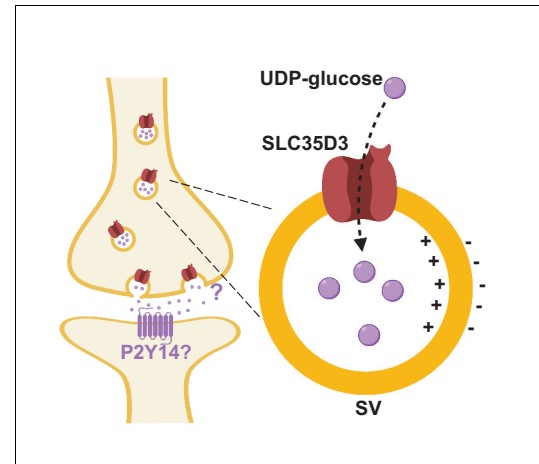

**Figure 6.** Working model depicting SLC35D3 as a putative UDP-glucose transporter on SVs. SLC35D3 is a vesicular transporter which potentially mediate transport of UDP-glucose into SVs. UDP-glucose may function as a signaling molecule through a GPCR namely P2Y14.

noise/signal-to-background ratio. Here in addition to function as an extracellular signaling molecule, UDP-glucose is also known to be accumulated in ER/Golgi for glycosylation of proteins (*Perez and Hirschberg, 1986*). This transport activity mediated by endogenous transporters contributes to the basal signal and limits the performance of overexpressed SLC35D3 in metabolic profiling as well as the transport assay based on organelles derived from the secretory pathways. Further optimization of the deorphanization strategy, e.g., knockingout endogenous transporters can be tested to maximize the signal-to-noise ratio.

SLC35D3 is expressed primarily in striatal neurons that project to the substantia nigra and the globus pallidus externa in the brain (*Lobo et al., 2006*), and mice with a recessive mutation in the *SLC35D3* gene have decreased motor activity, impaired energy expenditure, and develop obesity (*Zhang et al., 2014*). Thus, an important question for future studies is how SLC35D3 and its substrate UDP-glucose play a role in these circuits. The substrate of SLC35D3, UDP-glucose, is generally synthesized and exists in the cytoplasm (*Hirschberg et al., 1998*). In our hypothesis, UDP-glucose will be transported into SVs in SLC35D3-positive neurons and undergo regulated exocytosis upon stimulations. After the extracellular signaling process, UDP-glucose can be degraded by ecto-nucleotide pyrophosphatase/phosphodiesterases (E-NPPs), which is widely known to metabolize nucleoside triphosphates (*Lazarowski and Harden, 2015*).

Interestingly, previous studies regarding G protein–coupled receptors (GPCRs) found that UDP-sugars, including UDP-glucose, could serve as the ligand of the purinergic receptor P2Y14 (*Chambers et al., 2000*; *Freeman et al., 2001*), indicating that nucleotide sugars may function as extracellular signaling molecules, a notion supported by the fact that the P2Y14 receptor is widely expressed in a variety of brain regions and cell types (*Chambers et al., 2000*; *Lee et al., 2003*; *Zeisel et al., 2018*). The P2Y14 receptor is coupled primarily with the Gαi protein (*Chambers et al., 2000*; *Inoue et al., 2019*), which does not elicit an excitatory downstream calcium signal. Thus, whether the P2Y14 receptor plays a role in SLC35D3-expressing neurons is an interesting question that warrants investigation.

In addition to function in the central nervous system, it is possible that SLC35D3 also plays a role in the peripheral tissues. SLC35D3 can be localized to secretory organelles in platelets, and mutations on SLC35D3 lead to malfunction of the secretory organelles in platelets of mice, which resembles HPS syndrome that causes bleeding in humans (*Chintala et al., 2007*; *Meng et al., 2012*). Moreover, UDP-sugars can mediate vasoconstriction of the porcine coronary artery through the P2Y14 receptor (*Abbas et al., 2018*). Whether or not there is a general principle of vesicular UDP-glucose release mediated by SLC35D3 in different tissues would be an important question to answer.

# Materials and methods

## Key resources table

| Reagent type (species) or resource | Designation | Source or reference | Identifiers | Additional information |
|---|---|---|---|---|
| Gene (*Mus musculus*) | SYP | GenBank | NM_009305.2 | |
| Gene (*Homo sapiens*) | SLC35A2 | GenBank | NM_005660.3 | |
| Gene (*Mus musculus*) | SLC35D3 | GenBank | BC139194.1 | |
| Gene (*Homo sapiens*) | SLC35D3 | GenBank | KJ896073.1 | |
| Strain, strain background (*Mus musculus*) | Wild-type | Charles River | C57BL6/J, RRID:MGI:5650797 | |
| Strain, strain background (AAV) | AAV-PhP.eB hSyn-SLC35D3-EGFP-3xFlag | Vigene | | Titer: $7.68 \times 10^{13}$ gc/ml |
| Cell line (*Homo sapiens*) | HEK293T | ATCC | RRID:CVCL_0063 | |

*Continued on next page*

*Continued*

| Reagent type (species) or resource | Designation | Source or reference | Identifiers | Additional information |
|---|---|---|---|---|
| Cell line (*Homo sapiens*) | SLC35A2KO cell line | This paper | | Materials and methods section, 'KO cell line establishment and validation' |
| Antibody | Polyclonal rabbit anti-VGLUT1 | Synaptic Systems | Cat. #: 135302 RRID:AB_887877 | WB dilution 1:1000 |
| Antibody | Polyclonal rabbit anti-VGLUT2 | Synaptic Systems | Cat. #: 135402 RRID:AB_2187539 | WB dilution 1:1000 |
| Antibody | Monoclonal mouse anti-SYP | Synaptic Systems | Cat. #: 101011 RRID:AB_887824 | WB and IF dilution 1:1000 |
| Antibody | Polyclonal rabbit anti-SYP | Cell Signaling Technology | Cat. #: 5461 RRID:AB_10698743 | WB dilution 1:1000 |
| Antibody | Monoclonal mouse anti-VAMP2 | Synaptic Systems | Cat. #: 104211 RRID:AB_887811 | WB dilution 1:1000 |
| Antibody | Monoclonal mouse anti-PSD95 | NeuroMab | Cat. #: 75–028 RRID:AB_2292909 | WB dilution 1:1000 |
| Antibody | Monoclonal mouse anti-Flag | Sigma-Aldrich | Cat. #: F9291 RRID:AB_439698 | WB dilution 1:1000 |
| Antibody | Polyclonal chicken anti-GFP | Abcam | Cat. #: Ab13970 RRID:AB_300798 | IF dilution 1:1000 |
| Antibody | Monoclonal rabbit anti-CALR | Cell Signaling Technology | Cat. #: 12238 RRID:AB_2688013 | WB dilution 1:1000 |
| Antibody | Polyclonal rabbit anti-GM130 | Cell Signaling Technology | Cat. #: 12480 RRID:AB_2797933 | WB dilution 1:500 |
| Antibody | Polyclonal rabbit anti-Chg A | Synaptic Systems | Cat. #: 259003 RRID:AB_2619972 | WB and IF dilution 1:500 |
| Recombinant DNA reagent | pN3-human SLC35D3-mCherry (Plasmid) | This paper | | Materials and methods section, 'Molecular biology' |
| Recombinant DNA reagent | pN3-mouse SYP-EGFP (Plasmid) | This paper | | Materials and methods section, 'Molecular biology' |
| Recombinant DNA reagent | pN3-rat VGLUT1-APEX2 (Plasmid) | This paper | | Materials and methods section, 'Molecular biology' |
| Recombinant DNA reagent | pN3-OMM-APEX2 (Plasmid) | This paper | | Materials and methods section, 'Molecular biology' |
| Recombinant DNA reagent | pN3-human SLC35D3-APEX2 (Plasmid) | This paper | | Materials and methods section, 'Molecular biology' |
| Recombinant DNA reagent | pPacific-mouse SLC35D3-EGFP (Plasmid) | This paper | | Materials and methods section, 'Molecular biology' |
| Recombinant DNA reagent | pPacific-human SLC35D3-EGFP (Plasmid) | This paper | | Materials and methods section, 'Molecular biology' |
| Recombinant DNA reagent | pPacific-human SLC35A2-EGFP (Plasmid) | This paper | | Materials and methods section, 'Molecular biology' |
| Recombinant DNA reagent | pAAV-hSyn-human SLC35D3-EGFP-3xFlag (AAV vector) | This paper | | Materials and methods section, 'Molecular biology' |
| Recombinant DNA reagent | pLenti hSyn-human SLC35D3-EGFP-3xFlag (lenti vector) | This paper | | Materials and methods section, 'Molecular biology' |
| Recombinant DNA reagent | Human ORFeome 8.1 | Center for Cancer Systems Biology | http://horfdb.dfci.harvard.edu/ | Full-length human cDNAs |
| Recombinant DNA reagent | DNASU | NIGMS-funded Protein Structure Initiative (PSI) | https://dnasu.org/DNASU/Home.do | Full-length human cDNAs |

*Continued on next page*

*Continued*

| Reagent type (species) or resource | Designation | Source or reference | Identifiers | Additional information |
|---|---|---|---|---|
| Recombinant DNA reagent | The PlasmID Repository | DF/HCC DNA Resource Core at Harvard Medical School | https://plasmid.med.harvard.edu/PLASMID/Home.xhtml | Full-length human cDNAs |
| Chemical compound, drug | UDP-glucose | Santa Cruz | Cat. #: sc-296687 | |
| Chemical compound, drug | UDP-galactose | Santa Cruz | Cat. #: sc-286849A | |
| Chemical compound, drug | UDP-N-acetylgalactosamine | Sigma-Aldrich | Cat. #: U5252 | |
| Chemical compound, drug | UDP-N-acetylglucosamine | Sigma-Aldrich | Cat. #: U4375 | |
| Chemical compound, drug | UDP-xylose | SugarsTech | Cat. #: SN02004 | |
| Chemical compound, drug | UDP-glucuronic acid | Santa Cruz | Cat. #: sc-216043 | |
| Chemical compound, drug | CMP-sialic acid | Sigma-Aldrich | Cat. #: C8271 | |
| Chemical compound, drug | GDP-fucose | Santa Cruz | Cat. #: sc-221696A | |
| Chemical compound, drug | GDP-mannose | Santa Cruz | Cat. #: sc-285856A | |
| Chemical compound, drug | Uridine diphosphate glucose [6–3H] | PerkinElmer | Cat. #: NET1163250UC | |
| Chemical compound, drug | Uridine diphosphate galactose [1–3H] | ARC | Cat. #: ART0737 | |
| Chemical compound, drug | Uridine diphosphate N-acetylglucosamine [6–3H] | ARC | Cat. #: ART0128 | |
| Chemical compound, drug | Valinomycin | Sigma-Aldrich | Cat. #: V0627 | |
| Chemical compound, drug | Nigericin | Sigma-Aldrich | Cat. #: N7143 | |
| Chemical compound, drug | FCCP | Sigma-Aldrich | Cat. #: C2920 | |
| Chemical compound, drug | N-Ethylmaleimide | Sigma-Aldrich | Cat. #: E3876 | |
| Chemical compound, drug | Bafilomycin A1 | abcam | Cat. #: ab120497 | |
| Other | Protein G dynabeads | Thermo | Cat. #: 10004D | |

## Animals

Postnatal 0-day-old (P0) Sprague-Dawley rats (Beijing Vital River Laboratory) and adult (P42-56) wild-type C57BL/6J (Beijing Vital River Laboratory) were used in this study. All animals were raised in a temperature-controlled room with a 12 hr/12 hr light–dark cycle, and all animal procedures were performed using protocols approved by the Animal Care and Use Committees at Peking University.

## Molecular biology

DNA fragments were cloned using PCR amplification with primers (TsingKe Biological Technology) containing 30 bp of overlap. The fragments were then assembled into plasmids using Gibson assembly (*Gibson et al., 2009*). All plasmid sequences were verified using Sanger sequencing (TsingKe Biological Technology). For the localization studies in cultured neurons, the open-reading frames (e.g., SLC-mCherry, SLC-APEX2, SYP-EGFP, organelle marker-EGFP, etc.) were cloned into the N3 vector under the control of the CAG promoter. To generate stable cell lines expressing various SLC35 transporters, we generated the pPacific vector containing a 3' terminal repeat, the CAG promoter, a

P2A sequence, the puroR gene, and a 5' terminal repeat; the genes of interest were then cloned into a modified pPiggyBac (namely pPacific) vector using Gibson assembly. Two mutations (S103P and S509G) were introduced in pCS7-PiggyBAC (ViewSolid Biotech) to generate a hyperactive pig-gyBac transposase for generating the stable cell lines. For the AAV and lentivirus, hSyn-hSLC35D3-EGFP-3xFlag was cloned into pLenti and pAAV vectors independently.

## Lentiviral production

The lentivirus was produced by transfection of HEK-293T cells with the pLenti-hSyn-hSLC35D3-EGFP-3xFlag in combination with the VSV-G envelope and packaging plasmids. Twenty-four hours after transfection, the media was changed to fresh DMEM (Gibico) with 10% (v/v) fetal bovine serum (Gibco) and 1% penicillin-streptomycin (Gibco). Forty-eight hours after transfection, the virus containing supernatant was collected from the cells and centrifuged at 1000 g for 5 min to remove cells and debris. Supernatants were aliquoted and stored in −80℃.

## Preparation and fluorescence imaging of cultured cells

The HEK293T cell line is from ATCC. No mycoplasma contamination was detected. HEK293T cells were cultured at 37℃ in 5% $CO_2$ in DMEM (Gibco) supplemented with 10% (v/v) fetal bovine serum (Gibco) and 1% penicillin-streptomycin (Gibco). For transfection, cells in six-well plates were incubated in a mixture containing 1 μg DNA and 3 μg PEI for 6 hr, and fluorescence imaging was performed after the generation of a stable cell line.

Rat cortical neurons were prepared from P0 Sprague-Dawley rat pups (Beijing Vital River Laboratory). In brief, cortical neurons were dissociated from dissected rat brains in 0.25% trypsin-EDTA (GIBCO), plated on 12 mm glass coverslips coated with poly-D-lysine (Sigma-Aldrich), and cultured at 37℃ in 5% $CO_2$ in Neurobasal medium (Gibco) containing 2% B-27 supplement (Gibco), 1% Gluta-MAX (Gibco), and 1% penicillin-streptomycin (Gibco). After 7–9 days in culture, the neurons were transfected with SLC-mCherry, SYP-EGFP, organelle markers, or SLC-APEX2, and fluorescence imaging was performed 2–4 days after transfection. For AAV or lentivirus expressing epitope-tagged SLC35D3, neurons were infected after 7–9 days in culture, and fluorescence imaging was performed 4–7 days after infection.

Cultured cells were imaged using an inverted Ti-E A1 confocal microscope (Nikon) equipped with a 40×/1.35 NA oil-immersion objective, a 488 nm laser, and a 561 nm laser. During fluorescence imaging, the cells were either bathed or perfused in a chamber containing Tyrode's solution consisting of (in mM): 150 mM NaCl, 4 mM KCl, 2 mM $MgCl_2$, 2 mM $CaCl_2$, 10 mM HEPES, and 10 mM glucose (pH 7.4).

Localization imaging data of SLC-mCherry fluorescence overlapping with SYP-EGFP puncta were firstly manually selected by three researchers in a double-blind fashion. The selected SLC-mCherry images were further quantified to obtain a co-localization ratio with SYP-EGFP using the modified in silica Puncta Analyzer tool (see Source code file: in silica Puncta Analyzer tool), as described previously (*Kimura et al., 2007*). By using the plugin based on Image J software:

1. To adjust the contrast of raw green and red images, saturation was set to 0.35 with 25 min and 96 max. Then the images were processed with 'subtract background' and 'autothreshold'.
2. A colocalized channel of green and red channels was synthesized by a plug-in 'co-localization'.
3. Green and colocalized channels were transformed into binary images.
4. Synaptic boutons (puncta < 100 μm$^2$) from green and colocalized channels were extracted by 'analyze particles'.
5. Co-localization ratio = N (puncta in colocalized channel)/N (puncta in green channel).

## Immunostaining

Cells were firstly washed two times with PBS, followed by fixation in 4% PFA in PBS for 15 min, and then washed three times with PBS for 10 min each. Later, cells were permeabilized in 0.2% TritonX-100 in PBS for 20 min and were washed three times with PBS for 10 min each. After that, cells were blocked in 5% BSA in PBS for 1 hr. Primary antibodies were added to each coverslip: monoclonal mouse anti-SYP (101011; Synaptic Systems), polyclonal chicken anti-GFP (ab13970; Abcam), and polyclonal rabbit anti-Chg A (259003, Synaptic Systems). Cells were incubated overnight at 4℃. Following this, cells were washed three times with PBS for 10 min each. Secondary antibodies were

then added: goat anti-chicken Alexa Fluor 488, goat anti-mouse iFluor 555, and goat anti-rabbit iFluor 647. Cells were incubated at room temperature for 2 hr and washed three times with PBS for 10 min each. Cells were imaged by confocal microscopy as described above.

## Proteomics analysis of SVs

Thirty minutes prior to use, 5 µg of antibody was conjugated to 50 µl Protein G M-280 dynabeads at room temperature in KPBS buffer containing (in mM): 136 KCl and 10 $KH_2PO_4$ (pH 7.25). The brain was removed from an adult (P42-56) C57BL/6J mouse, homogenized using a ball-bearing homogenizer (10 µm clearance) in 3 ml ice-cold KPBS, and centrifuged at 30,000 g for 20 min. The supernatant (input) containing the SVs was collected and incubated with antibody-conjugated dynabeads for 1 hr at 0°C for immunoisolation. Dynabead-bound SVs were washed three times with KPBS and eluted by incubating the samples with SDS–PAGE sample loading buffer. The SV samples were heated for 10 min at 70°C and centrifuged for 2 min at 14,000 rpm, and the supernatants were transferred to clean tubes. The protein samples were then subjected to SDS–PAGE for western blotting and HPLC–MS, respectively.

The resolved proteins in SDS–PAGE were digested and extracted from the gel pieces using acetonitrile containing 0.1% formic acid (FA). The samples were then dried in a vacuum centrifuge concentrator at 30°C and resuspended in 10 µl 0.1% FA.

Using an Easy-nLC 1200 system, 5 µl of sample was loaded at a rate of 0.3 µl/min in 0.1% FA onto a trap column (C18, Acclaim PepMap 100 75 µm $\times$ 2 cm; Thermo Fisher Scientific) and eluted across a fritless analytical resolving column (C18, Acclaim PepMap 75 µm $\times$ 15 cm; Thermo Fisher Scientific) with a 75 min gradient of 4–30% LC–MS buffer B at 300 nl/min; buffer A contained 0.1% FA, and buffer B contained 0.1% FA and 80% acetonitrile.

The peptides were directly injected into an Orbitrap Fusion Lumos (Thermo Fisher Scientific) using a nano-electrospray ion source with an electrospray voltage of 2.2 kV. Full-scan MS spectra were acquired using the Orbitrap mass analyzer (m/z range: 300–1500 Da), with the resolution set to 60,000 (full width at half maximum or FWHM) at m/z = 200 Da. Full-scan target was 5e5 with a maximum fill time of 50 ms. All data were acquired in profile mode using positive polarity. MS/MS spectra data were acquired using Orbitrap with a resolution of 15,000 (FWHM) at m/z = 200 Da and higher-collisional dissociation (HCD) MS/MS fragmentation. The isolation width was 1.6 m/z.

## Intravenous injection

The procedure was adapted from previous study (*Challis et al., 2019*). Briefly, WT female adult (P42–48) C57BL/6N mice were anesthetized by an intraperitoneal (i.p.) injection of 2,2,2-tribromoethanol (Avertin, 500 mg/kg body weight, Sigma-Aldrich). AAV-PhP.eB was delivered by retro-orbital injection to the mice at 5 $\times$ $10^{11}$ genome copy (gc), and western blot analysis was conducted 3 weeks after injection.

## Western blot

Protein lysates were denatured by the addition of 2$\times$ sample buffer followed by 70°C treatment for 10 min. Samples were resolved by 10% SDS–PAGE, transferred for 1 hr at room temperature at 25 V to NC membranes, and analyzed by immunoblotting. Membranes were firstly stained by Ponceau S staining followed by washing with TBST and blocking with 5% non-fat milk prepared in TBST for 1 hr at room temperature. Membranes were then incubated with primary antibodies in 5% non-fat milk TBST overnight at 4°C, followed by washing with TBST three times, 10 min each. Membranes were incubated with the corresponding secondary antibodies in 5% non-fat milk for 2 hr at room temperature. Membranes were then washed three more times, 10 min each, with TBST before being visualized using chemiluminescence. Antibodies used were polyclonal rabbit anti-VGLUT1 (135302; Synaptic Systems), polyclonal rabbit anti-VGLUT2 (135402; Synaptic Systems), monoclonal mouse anti-SYP (101011; Synaptic Systems), polyclonal rabbit anti-SYP (5461; Cell Signaling Technology), monoclonal mouse anti-VAMP2 (104211; Synaptic Systems), monoclonal mouse anti-PSD95 (75-028; NeuroMab), monoclonal mouse anti-Flag (F9291; Sigma-Aldrich), monoclonal rabbit anti-CALR (12238, Cell Signaling Technology), polyclonal rabbit anti-GM130 (12480, Cell Signaling Technology), and polyclonal rabbit anti-Chg A (259003, Synaptic Systems).

## Electron microscopy

Antibody-conjugated dynabeads were pelleted by centrifugation and subsequently resuspended in 1.5% agarose in 0.1 M phosphate buffer (PB, pH 7.4). Small agarose blocks were cut out, fixed overnight at 4℃ using 4% glutaraldehyde in 0.1 M PB at pH 7.4, followed by post-fixation of 1% osmium tetroxide for 1 hr and treatment of 0.25% uranyl acetate overnight at 4℃. The samples were then dehydrated in a graded ethanol series (20%, 50%, 70%, 80%, 90%, 95%, 100%, 100%) at 8 min per step and then changed to propylene oxide for 10 min. The cells were then infiltrated in Epon 812 resin using a 1:1 ratio of propylene oxide and resin for 4 hr, followed by 100% resin twice at 4 hr each; finally, the beads were placed in fresh resin and polymerized in a vacuum oven at 65℃ for 24 hr. After polymerization, ultrathin sections were cut and stained with lead citrate.

For APEX2-based EM labeling, the procedure was adapted from previous study (*Martell et al., 2012*). Transfected neurons were firstly fixed with 2% glutaraldehyde in 0.1 M PB at room temperature, quickly placed on ice, and incubated on ice for 45–60 min. The cells were rinsed with chilled PB twice at 5 min each before adding 20 mM glycine to quench any unreacted glutaraldehyde. The cells were then rinsed three times with PB at 5 min each. Freshly prepared solution containing 0.5 mg/ml 3,3′-diaminobenzidine (DAB) tetrahydrochloride and 10 mM $H_2O_2$ was then added to the cells. After 5–10 min, the reaction was stopped by removing the DAB solution and rinsing three times with chilled PB at 5 min each. The cells were then incubated in 2% osmium tetroxide in 0.1 M PB combined with 0.1 M imidazole (pH 8.0) for 30 min in a light-proof box. The cells were then rinsed six times with water at 5 min each and then incubated in 2% (w/v) aqueous uranyl acetate overnight at 4℃. The cells were rinsed six times with water at 5 min each, then dehydrated in a graded ethanol series (20%, 50%, 70%, 80%, 90%, 95%, 100%, 100%) at 8 min per step, and then rinsed once in anhydrous ethanol at room temperature. The cells were then infiltrated in Epon 812 resin using a 1:1, 1:2, and 1:3 (v/v) ratio of anhydrous ethanol and resin for 1 hr, 2 hr, and 4 hr, respectively, followed by 100% resin twice at 4 hr each; finally, the cells were placed in fresh resin and polymerized in a vacuum oven at 65℃ for 24 hr.

The embedded cells were cut into 60 nm ultrathin sections using a diamond knife and imaged using a FEI-Tecnai G2 20 TWIN transmission electron microscope operated at 120 kV.

## KO cell line establishment and validation

The SLC35A2KO cell line was constructed by transient co-transfection of plasmids expressing mCherry and sgRNAs targeting the SLC35A2 gene, and a plasmid expressing spCas9. The sgRNA sequences were as follows: atgccaacatggcagcggtt, ggtggttccaccgcggcgcc, ggcggtttccgcgggtgcat, and gactgtctcacccgcactgg. Single cells with high mCherry signal were sorted and seeded in 96-well plates 1 week after transfection. After cell expansion, the SLC35A2KO DNA fragments of target loci were independently amplified by PCR with a primer pair (SLC35A2seqF: tttaggagcggaggagaaaag; SLC35A2seqR: ctctcagaatgttctcttcccc). The purified PCR products were sequenced, and the insertions and deletions (indels) within the SLC35A2 gene caused by sgRNA/Cas9 were analyzed with an online tool (http://crispid.gbiomed.kuleuven.be/) (*Dehairs et al., 2016*). Functional validation was done by radioactive transport assay.

## Organelle fractionation

Stable cell lines grown in two 15 cm dishes were washed twice with either ice-cold KPBS (for metabolite detection) or sucrose buffer containing 0.32 M sucrose and 4 mM HEPES-NaOH (pH 7.4) (for the uptake assay), and then gently scraped and collected into 1 ml of the corresponding buffer. The cells were then homogenized using a ball-bearing homogenizer (10 μm clearance). The homogenate was centrifuged at 13,000 g for 10 min to remove the nuclei and cellular debris. The resulting supernatant was centrifuged at 200,000 g for 25 min. For metabolite profiling, the pellet was washed three times in ice-cold KPBS, and the metabolites were extracted in 80% methanol, freeze-dried, and stored at −80℃. For the transport assay, the pellet was resuspended in uptake assay buffer containing 0.32 M sucrose, 2 mM KCl, 2 mM NaCl, 4 mM $MgSO_4$, and 10 mM HEPES-KOH (pH 7.4), aliquoted, and stored at −80℃.

For SV fractionation, the procedure was adapted from previous study (*Huttner et al., 1983*) (see also *Figure 2—figure supplement 2*). Briefly, mouse brains were gently homogenized in sucrose buffer containing 0.32 M sucrose and 4 mM HEPES-NaOH (pH 7.4) on ice. The homogenate was

centrifuged at 800 g for 10 min to remove the nuclei and cellular debris. The resulting supernatant (S1) was collected and centrifuged at 9200 g for 15 min. The pellet (P2) was resuspended in sucrose buffer and recentrifuged at 10,200 g for 15 min. The resulting pellet (P2') was resuspended in 1 ml sucrose buffer and then added with 9 ml ice-cold water. After three strokes, the lysate was immediately added with 80 µl 1M HEPES-NaOH buffer (pH 7.4) and kept on ice for 30 min. The lysate was then centrifuged at 25,000 g for 20 min. The resulting supernatant (LS1) was further centrifuged at 165,000 g for 2 hr to get a pellet (LP2) of crude SVs.

## Targeted metabolite profiling

Samples were analyzed using a TSQ Quantiva Ultra triple-quadrupole mass spectrometer coupled with an Ultimate 3000 UPLC system (Thermo Fisher Scientific) equipped with a heated electrospray ionization probe. Chromatographic separation was achieved using gradient elution on a Hypercarb PGC column (2.1 × 100 mm, 1.7 µm, Thermo Fisher Scientific). Mobile phase A consisted of 5 mM ammonium bicarbonate dissolved in pure water, and mobile phase B consisted of 100% acetonitrile. A 25 min gradient with a flow rate of 250 µl/min was applied as follows: 0–1.2 min, 4% B; 1.2–19 min, 4–35% B; 19–20 min, 35–98% B; 20–22 min, 98% B; 22–25 min 4% B. The column chamber and sample tray were kept at 45℃ and 10℃, respectively. Data were acquired using selected reaction monitoring in negative switch ion mode, and optimal transitions are reported as the reference. Both the precursor and fragment ion fractions were collected at a resolution of 0.7 FWHM. The source parameters were as follows: spray voltage: 3000 V; ion transfer tube temperature: 350℃; vaporizer temperature: 300℃; sheath gas flow rate: 35 arbitrary units; auxiliary gas flow rate: 12 arbitrary units; collision-induced dissociation gas pressure: 1.5 mTorr.

## Uptake assay

For the radiolabeled substrate transport assay, 20 µg of the membrane fraction was incubated with the indicated concentration of radiolabeled substrate at 37℃ for 5 min (unless otherwise). The reaction was terminated using the same volume of ice-cold assay buffer. The samples were then trapped on a 0.7 µm GF/F glass fiber filter (Whatman) and washed twice. The radioactivity retained on the filter was measured using liquid scintillation.

## Quantification and statistical analysis

Imaging data from cultured cells were processed using ImageJ software (NIH). SV proteomics data were analyzed using MaxQuant_1.6.10.43 (MPI). The metabolite profiling data were analyzed and quantified using Xcalibur version 3.0.63 (Thermo Fisher Scientific). Sequence data for generating the phylogenic tree of were analyzed by MEGA-X. Cartoons created using BioRender (http://www.biorender.com). All summary data are presented as the mean ± s.e.m., and group data were compared using the Student's t-test or the Kruskal–Wallis ANOVA test; *p<0.05, **p<0.01, ***p<0.001, and n. s., not significant (p>0.05).

## Acknowledgements

This work was supported by the Beijing Municipal Science and Technology Commission (Z181100001318002), grants from the Peking-Tsinghua Center for Life Sciences, and grants from the State Key Laboratory of Membrane Biology at Peking University School of Life Sciences.

We thank the Proteomics Core in National Center for Protein Sciences at Peking University, particularly Dong Liu for providing technical assistance. We thank Shitang Huang for helping with the radioactive transport assay. We also thank the Metabolomics Facility at Technology Center for Protein Sciences in Tsinghua University, particularly Xueying Wang and Li'na Xu, for providing technical assistance.

## Additional information

### Funding

| Funder | Grant reference number | Author |
|---|---|---|
| Beijing Municipal Science and Technology Commission | Z181100001318002 | Yulong Li |
| Center for Life Sciences | | Yulong Li |
| State Key Laboratory of Membrane Biology | | Yulong Li |

The funders had no role in study design, data collection and interpretation, or the decision to submit the work for publication.

### Author contributions

Cheng Qian, Conceptualization, Resources, Data curation, Validation, Investigation, Methodology, Writing - original draft, Writing - review and editing; Zhaofa Wu, Rongbo Sun, Data curation, Formal analysis, Validation, Investigation, Methodology, Writing - review and editing; Huasheng Yu, Data curation, Validation, Investigation, Writing - review and editing; Jianzhi Zeng, Data curation, Formal analysis, Investigation, Methodology, Writing - review and editing; Yi Rao, Conceptualization, Supervision, Writing - review and editing; Yulong Li, Conceptualization, Supervision, Funding acquisition, Writing - original draft, Writing - review and editing

### Author ORCIDs

Cheng Qian (ID) https://orcid.org/0000-0002-9926-4406
Zhaofa Wu (ID) https://orcid.org/0000-0003-2027-5194
Rongbo Sun (ID) https://orcid.org/0000-0002-4364-8435
Huasheng Yu (ID) https://orcid.org/0000-0001-5641-2512
Jianzhi Zeng (ID) https://orcid.org/0000-0002-5380-6281
Yi Rao (ID) http://orcid.org/0000-0002-0405-5426
Yulong Li (ID) https://orcid.org/0000-0002-9166-9919

### Ethics

Animal experimentation: All animal procedures were performed using protocols approved by the Institutional Animal Care and Use Committee at Peking University. ( LSC LiYL 1 ).

### Decision letter and Author response

Decision letter https://doi.org/10.7554/eLife.65417.sa1
Author response https://doi.org/10.7554/eLife.65417.sa2

## Additional files

### Supplementary files

- Source code 1. in silica Puncta Analyzer tool.
- Supplementary file 1. Vesicular transporters identified in SLC localization profiling.
- Supplementary file 2. SLC transporters enriched in immunoisolated synaptic vesicles.
- Transparent reporting form

### Data availability

All data generated or analysed during this study are included in the manuscript and supporting files.

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
