## [Decision Letter]

**Acceptance summary:**

This paper raises the interesting possibility that an orphan transporter related to nucleotide-sugar transporters, packages UDP-glucose into synaptic vesicles, thereby conferring the potential for UDP-glucose to serve as an extracellular signal. The authors present a novel workflow to identify potential vesicular transporters that importantly does not rely solely on the low throughput method of randomly testing candidate substrates. The paper thus will be of potential interest to researchers who study neurochemistry and vesicular physiology.

**Decision letter after peer review:**

Thank you for submitting your article "Localization, proteomics, and metabolite profiling reveal a putative vesicular transporter for UDP-glucose" for consideration by *eLife*. Your article has been reviewed by 4 peer reviewers, one of whom is a member of our Board of Reviewing Editors, and the evaluation has been overseen Olga Boudker as the Senior Editor. The following individual involved in review of your submission has agreed to reveal their identity: David E. Krantz (Reviewer #4).

The reviewers have discussed the reviews with one another and the Reviewing Editor has drafted this decision to help you prepare a revised submission.

Summary:

The manuscript by Qian et al., reports SLC35D3 as a synaptic vesicle transporter for UDP-glucose. This transporter has been implicated basal ganglia function, energy expenditure and obesity. The manuscript is well-written and the study largely well-executed. The authors performed two screens to identify synaptic vesicle transporters, a localization assay in transfected rat neurons and a proteomic screen of immuno-isolated synaptic vesicle proteins in mice. From these screens they reported the identification of 40 and 20 transporters, respectively, and of these, 7 were present in both screens. Four are vesicular transporters with already known substrates and three are orphan transporters. The authors used metabolite profiling to identify UDP-glucose as the putative substrate of one of the identified orphan transporters, SLC35D3. Transport assays on isolated membranes from transfected HEK293 cells were then used to characterize further its substrates and inhibitors and dependence on the proton electrochemical gradient. While these experiments point to SLC35D3 as a novel synaptic vesicle transporter, which would be the important finding of this study, there are a number of concerns raised by the reviewers as to the veracity of this conclusion that will first need to be addressed, as outlined below.

Essential Revisions:

1. There is a concern that the localization screen and the EM resulted in over-expression of the transporter, which could cause its mis-localization to synapses. If there is an available antibody that the authors can show is specific to the transporter, the authors could immunostain brain slices or cultured neurons to show co-localization of the endogenous transporter with synaptophysin. If an antibody is not available, the authors could use lentivirus with an epitope tagged transporter to try to avoid over-expression. Since synaptophysin also localizes to secretory granules, it would be helpful to co-stain with a bona fide secretory granule-specific marker such as Chromogranin A but also see #2 below.

2. There is a concern that the synaptic vesicle isolation is not pure enough to conclude that SLC35D3 is on synaptic vesicles (as opposed to secretory granules or other intracellular organelles). To mitigate this, the authors could perform subcellular fractionation to isolate synaptic vesicles, secretory granules, ER, Golgi etc by size and relative density differences via a classical biochemical method such as density gradient centrifugation. This would allow the authors to determine the relative enrichment of the transporter in these compartments which is also important to know since SLC35 family members have been reported to reside in Golgi/ER compartments.

3. There are also concerns regarding the dependence on the proton electrochemical gradient. The authors show that BafA1, which inhibits the proton ATPase does not have an effect on transport. This contradicts their findings with FCCP and nigericin. The authors should discuss the finding with BafA1 relative to the other findings and whether the transporter acts as an obligate exchanger. Related to the possibility that the transporter instead resides on Golgi/ER, the authors should compare the activity of SLC35D3 with a more canonical ER/Golgi member of the family to demonstrate the difference in mechanism. This may also involve fractionation of the membranes as mentioned in #2.

[Editors' note: further revisions were suggested prior to acceptance, as described below.]

Thank you for resubmitting your work entitled "Localization, proteomics, and metabolite profiling reveal a putative vesicular transporter for UDP-glucose" for further consideration by *eLife*. Your revised article has been reviewed by 4 peer reviewers, one of whom is a member of our Board of Reviewing Editors, and the evaluation has been overseen by a Senior Editor, Olga Boudker.

The manuscript has been improved but there are some remaining issues that need to be addressed, as outlined below:

Essential revisions:

There is still some concern amongst the reviewers that the authors have not convincingly demonstrated that the transporter resides exclusively or primarily on synaptic vesicles in vivo. In the absence of an antibody that recognizes the transporter in brain slices, the reviewers suggested that the authors use lentivirus to minimize overexpression, which the authors did in cultured neurons and indeed expression was lower with the lentivirus and still colocalized to a significant extent with an antibody to synaptophysin. However, for the brain tissue subcellular fractionation study, the authors instead used AAV PhP.eB, which showed similar overexpression as plasmid transfection in cultured cells raising some doubt about its subcellular distribution in this context. Also, although the fractionation did show some enrichment on LP2, the levels of the transporter were otherwise fairly similar across fractions. Lastly, the LP2 fraction is too crude as localization of the transporter to other organelles within this fraction is instead possible.

Thus, the reviewers request that the authors use density gradient fractionation to identify more precisely the identity of the organelle(s) where the transporter resides in vivo. (And lentivirus is preferred if possible over PhP.eB.) If this experiment is not included in the revised manuscript, the authors need to discuss the above mentioned limitations (use of PhP.eB virus likely leads to transporter overexpression in brain and interpretation of LP2 as being crude) when describing the fractionation experiment and the authors will need to temper their claim throughout the manuscript that the transporter localizes exclusively or primarily to SVs in vivo.

*Reviewer #1 (Recommendations for the authors):*

It would have been helpful to see what happens if the authors had injected the lentivirus construct into the mice and looked for colocalization with markers of synaptic vesicles (VGLUT1, VGAT, synaptophysin) as well as performed the fractionation study. It is likely they chose the PHP.eb because of the retro-orbital injection and supposedly good spread of the virus.

Otherwise the authors have satisfied my concerns.

*Reviewer #2 (Recommendations for the authors):*

Li and colleagues have provided a number of additional experiments and have satisfactorily addressed most of the points. However, one prominent concern remains regarding the subcellular localization of SLC35D3. Essentially, the authors performed a crude synaptosomal preparation and examined a fraction that is enriched in synaptic vesicles and has relatively little ER and Golgi. Nevertheless, the authors overstate the purity of the fraction examined. Indeed, as much as ~30% of SLC35D3 co-localizes with chromogranin A, a canonical secretory granule marker suggesting that this fraction is enriched in secretory granules as well as synaptic vesicles. One strong possibility therefore is that SLC35D3 is not purely found in synaptic vesicles but that it is also localized to other vesicular compartments including secretory granules. The authors did not test enriched the crude synaptic vesicle fractions like LP2 for other compartments like endoscopes, lysosomes etc. raising further questions about the definitive intracellular localization. This would also be in line with the discrepancy concerning the absence of being reported in recent synaptic vesicle proteomics studies such as Taoufiq et al. (2020). Therefore, without obtaining a highly purified synaptic vesicle fraction either through classical biochemical means or even via recently reported immuno-isolation (for example, see Chantranupong et al., *eLife* 2020), concluding that SLC35D3 is truly a synaptic vesicle transporter cannot be made conclusively.

*Reviewer #3 (Recommendations for the authors):*

The authors have responded very thoughtfully to the concerns raised. Questions still persist about the synaptic vesicle localization (still reliant on over-expression and the enrichment by fractionation is not strong), but there is not much more the authors can do given the low expression of endogenous 35D3. Similarly, the lack of bafilomycin effect is surprising since this usually results in run-down of the membrane potential--although the pH gradient may be more resistant (H^+^ efflux requires movement of a counterion). Since other family members differ in mechanism, I suspect this transporter does not depend on a H^+^ electrochemical gradient, accounting for the bafilomycin result, but it is difficult to explain the effect of nigericin as indirect, so again, there is not much more the authors can do and their comparison to another family member here is helpful. Again, it is unclear why the published proteomics has not identified 35D3 but the authors have done all they can and the results presented are reasonable.

*Reviewer #4 (Recommendations for the authors):*

The authors have done a thorough job responding to the reviewers' comments.

---

## [Author Response]

Essential Revisions:1. There is a concern that the localization screen and the EM resulted in over-expression of the transporter, which could cause its mis-localization to synapses. If there is an available antibody that the authors can show is specific to the transporter, the authors could immunostain brain slices or cultured neurons to show co-localization of the endogenous transporter with synaptophysin. If an antibody is not available, the authors could use lentivirus with an epitope tagged transporter to try to avoid over-expression. Since synaptophysin also localizes to secretory granules, it would be helpful to co-stain with a bona fide secretory granule-specific marker such as Chromogranin A but also see #2 below.

The reviewer raised an important concern, and we thank the reviewers for the suggestions. We have tested two customized antibodies against SLC35D3. Unfortunately, they could not efficiently stain overexpressed SLC35D3 in the immunostaining of cultured neurons (Author response image 1) or western blot analysis of HEK293T cells (Author response image 1), indicating the antibodies are less likely to recognize endogenous SLC35D3.

**Author response image 1. sa2fig1:** Test of anti-SLC35D3 antibodies (A) Representative images of cultured neurons expressing SYP-EGFP (green) or SLC35D3-mCherry (red) immunostained with anti-EGFP or anti-SLC35D3 antibodies (magenta), respectively. Arrowheads indicate SYP-EGFP positive puncta and arrows indicate SLC35D3 positive puncta. Scale bars: 10 μm. (B) Left: Representative images of HEK293T cells with/without transfection of SLC35D3-EGFP. Scale bars: 25 μm. Right: Western blot analysis of overexpressed SLC35D3-EGFP from lysed HEK293T cells. β-actin serves as an internal control.

As the reviewer suggested, we have packaged both lentivirus and AAV PhP.eB virus to ameliorate the expression level of SLC35D3 in neurons. The lowest expression level of epitope-tagged SLC35D3 was achieved using lentivirus, which was ~ 40% compared with plasmid transfection (Figure 1—figure supplement 1A,B). Then we focused on the localization of SLC35D3 in the lentivirus infected neurons (Figure 1—figure supplement 1C). The colocalization ratio between lentivirus-expressed SLC35D3 and SYP (SV marker) was ~60%, which is similar to that in the plasmid transfected neurons (~70%). Conversely, the colocalization ratio between SLC35D3 and Chg A (secretory granule marker) was ~30%. Thus, SLC35D3 with a relatively low expression level is mainly localized to SVs as opposed to secretory granules.

We have added this information into the revised manuscript:

Line 135-144:

“To avoid mis-localization caused by overexpression, we tested different delivery strategies for a low expression level on one candidate SLC35D3. The lowest expression level of epitope-tagged SLC35D3 was achieved using lentivirus, which was ~ 40% compared with plasmid transfection (Figure 1—figure supplement 1A,B). Then we focused on the localization of SLC35D3 in the lentivirus infected neurons (Figure 1—figure supplement 1C). The colocalization ratio between SLC35D3 and SYP (SV marker) was ~60%, which is similar to that in the plasmid transfected neurons (~70%). Given SYP may also be localized to secretory granules, we co-immunostained a secretory granule marker Chg A and found that the colocalization ratio between SLC35D3 and Chg A was ~30%. Taken together, SLC35D3 with relatively low expression level is mainly localized to synaptic vesicles as opposed to secretory granules.”

2. There is a concern that the synaptic vesicle isolation is not pure enough to conclude that SLC35D3 is on synaptic vesicles (as opposed to secretory granules or other intracellular organelles). To mitigate this, the authors could perform subcellular fractionation to isolate synaptic vesicles, secretory granules, ER, Golgi etc by size and relative density differences via a classical biochemical method such as density gradient centrifugation. This would allow the authors to determine the relative enrichment of the transporter in these compartments which is also important to know since SLC35 family members have been reported to reside in Golgi/ER compartments.

To address the reviewer’s concerns, we performed differential centrifugation [1] (Figure 2—figure supplement 2A) to fractionate the mouse brain for different subcellular components, and then determined the level of SLC35D3. Firstly, we conducted retro-orbital injection of AAV-PhP.eB virus to infect the mouse brain [2] (for enough purification input). The expression of epitope-tagged SLC35D3 was detected three weeks after AAV injection (Figure 2—figure supplement 2B). With the progress of differential centrifugation, we observed enrichment of SLC35D3 from P2’ (crude synaptosome) to LP2 (crude SVs) fraction, which was similar to known SV markers VGLUT1 and SYP. In contrast, the secretory granule marker Chg A, organelle markers of ER and Golgi were mainly enriched before P2’ (Figure 2—figure supplement 2C). SLC35D3 and VGLUT1 also appeared in P1 and S1 fractions, likely due to the reason that these membrane proteins are being produced and processed through the secretory pathway. In summary, our newly obtained subcellular fractionation data further validated that SLC35D3 is mainly on SVs.

We have added this information into the revised manuscript:

Line 184-195:

“To further dissect the subcellular distribution of one novel vesicular transporter candidate, SLC35D3, in different organelles, we performed differential centrifugation to fractionate the mouse brain [1] (Figure 2—figure supplement 2A). Firstly, we conducted retro-orbital injection of AAV-PhP.eB virus to infect the mouse brain [2] (for enough purification input). The expression of epitope-tagged SLC35D3 was detected three weeks after AAV injection (Figure 2—figure supplement 2B). With the progress of differential centrifugation, we observed enrichment of SLC35D3 from P2’ (crude synaptosome) to LP2 (crude SVs) fraction, which is similar to known SV markers VGLUT1 and SYP. In contrast, the secretory granule marker Chg A, organelle markers of ER and Golgi are majorly enriched before P2’ (Figure 2—figure supplement 2C). SLC35D3 and VGLUT1 also appeared in P1 and S1 fractions, likely due to the reason that these membrane proteins are being produced and processed through the secretory pathway. In summary, these data further validated that SLC35D3 is mainly on SVs.”

[1] Huttner, W. B., Schiebler, W., Greengard, P., and De Camilli, P. (1983). Synapsin I (protein I), a nerve terminal-specific phosphoprotein. III. Its association with synaptic vesicles studied in a highly purified synaptic vesicle preparation. The Journal of cell biology, 96(5), 1374-1388.

[2] Challis, R. C., Kumar, S. R., Chan, K. Y., Challis, C., Beadle, K., Jang, M. J., … and Gradinaru, V. (2019). Systemic AAV vectors for widespread and targeted gene delivery in rodents. Nature protocols, 14(2), 379-414.

3. There are also concerns regarding the dependence on the proton electrochemical gradient. The authors show that BafA1, which inhibits the proton ATPase does not have an effect on transport. This contradicts their findings with FCCP and nigericin. The authors should discuss the finding with BafA1 relative to the other findings and whether the transporter acts as an obligate exchanger. Related to the possibility that the transporter instead resides on Golgi/ER, the authors should compare the activity of SLC35D3 with a more canonical ER/Golgi member of the family to demonstrate the difference in mechanism. This may also involve fractionation of the membranes as mentioned in #2.

We thank the reviewers for pointing this out.

The inhibition mechanism of BafA1 is different from FCCP and nigericin. BafA1 directly inhibits the V-ATPase [1], thus indirectly affects the maintenance of proton electrochemical gradient across the SV membrane. SVs can still preserve proton electrochemical gradient after acute application of BafA1, as indicated by previous work using pH-dependent quantum dots to study SV Kiss and Run (KandR) and full-collapse fusion (FCF) [2]. In contrast, proton uncouplers including FCCP and nigericin can directly abolish the proton electrochemical gradient. There is a possibility that preserved proton electrochemical gradient after the acute application of BafA1 could support the transport of UDP-glucose by SLC35D3.

We thank the reviewers for raising a good question about the obligate exchanger activity. Till now we don’t know if SLC35D3 has this kind of transport activity. We have added a discussion about this into the revised manuscript, please see below.

To compare the transport mechanism of SLC35D3 with a canonical ER/Golgi localized SLC35 transporter, we investigated pharmacological properties of SLC35A3, which is an ER/Golgi localized UDP-N-Acetyl-glucosamine transporter [3] (Figure 5—figure supplement 1A). We found the pharmacological treatment including proton uncouplers didn’t significantly inhibit UDP-N-Acetyl-glucosamine transport, indicating SLC35A3 may have a different transport mechanism compared with SLC35D3 (Figure 5—figure supplement 1B). Moreover, the transport activity of GDP-mannose by a yeast homolog of the nucleotide-sugar transporters was neither sensitive to CCCP nor valinomycin [4], which also suggested different transport mechanisms among nucleotide-sugar transporters. Further studies by proteoliposome reconstitution of purified SLC35D3 can help to illustrate the detailed transport mechanism.

To clarify this, we have revised the manuscript accordingly:

Line 279-295:

“Unlike the proton uncouplers that directly abolish the proton electrochemical gradient, Bafilomycin A1 inhibits V-ATPase that indirectly affects the maintenance of proton electrochemical gradient [1]. […] Further studies by proteoliposome reconstitution of purified SLC35D3 can help to illustrate the detailed transport mechanism, e.g. if SLC35D3 has the obligate exchanger activity.”

[1] Yoshimori, T., Yamamoto, A., Moriyama, Y., Futai, M., and Tashiro, Y. (1991). Bafilomycin A1, a specific inhibitor of vacuolar-type H (+)-ATPase, inhibits acidification and protein degradation in lysosomes of cultured cells. Journal of Biological Chemistry, 266(26), 17707-17712.

[2] Zhang, Q., Li, Y., and Tsien, R. W. (2009). The dynamic control of kiss-and-run and vesicular reuse probed with single nanoparticles. Science, 323(5920), 1448-1453.

[3] Ishida, N., Yoshioka, S., Chiba, Y., Takeuchi, M., and Kawakita, M. (1999). Molecular cloning and functional expression of the human Golgi UDP-N-acetylglucosamine transporter. The Journal of Biochemistry, 126(1), 68-77.

[4] Parker, J. L., and Newstead, S. (2017). Structural basis of nucleotide sugar transport across the Golgi membrane. Nature, 551(7681), 521-524.

[Editors' note: further revisions were suggested prior to acceptance, as described below.]

Essential revisions:There is still some concern amongst the reviewers that the authors have not convincingly demonstrated that the transporter resides exclusively or primarily on synaptic vesicles in vivo. In the absence of an antibody that recognizes the transporter in brain slices, the reviewers suggested that the authors use lentivirus to minimize overexpression, which the authors did in cultured neurons and indeed expression was lower with the lentivirus and still colocalized to a significant extent with an antibody to synaptophysin. However, for the brain tissue subcellular fractionation study, the authors instead used AAV PhP.eB, which showed similar overexpression as plasmid transfection in cultured cells raising some doubt about its subcellular distribution in this context. Also, although the fractionation did show some enrichment on LP2, the levels of the transporter were otherwise fairly similar across fractions. Lastly, the LP2 fraction is too crude as localization of the transporter to other organelles within this fraction is instead possible.Thus, the reviewers request that the authors use density gradient fractionation to identify more precisely the identity of the organelle(s) where the transporter resides in vivo. (And lentivirus is preferred if possible over PhP.eB.) If this experiment is not included in the revised manuscript, the authors need to discuss the above mentioned limitations (use of PhP.eB virus likely leads to transporter overexpression in brain and interpretation of LP2 as being crude) when describing the fractionation experiment and the authors will need to temper their claim throughout the manuscript that the transporter localizes exclusively or primarily to SVs in vivo.

We thank the reviewers for the suggestions, and we would like to discuss the mentioned limitations.

Firstly, there is a trade-off in choosing the virus and delivery strategy. Even though lentivirus can ameliorate the expression level of protein of interest, it is difficult to achieve enough infection for biochemical fractionation study after injected into adult mouse brains since it can be only diffused and expressed in a limited brain region [1-3]. Injection into embryonic mouse brains was reported to result in a wide distribution after the developmental process [4], but it may take a few months which would be beyond the time scale of the revision. Therefore, we previously chose AAV-PhP.eB which could cross the blood-brain barrier and get delivered to the whole mouse brain easily [5] to achieve satisfactory starting input of biochemical experiments.

Secondly, using AAV-PhP.eB has its limitation in this case. The relatively high expression level of the transporter by AAV-PhP.eB may result in excess distribution of the protein in the secretory pathway, which subsequently affects the performance of either differential fractionation or density gradient fractionation for identifying the subcellular distribution of the protein of interest. This may also help to explain that the enrichment of SLC35D3 in LP2 is not as significant as SYP.

Thirdly, we agree that LP2 is crude which may contain other organelles so that we cannot rule out the possibility that SLC35D3 also localizes to other organelles such as secretory granules.

We have now added a paragraph, as shown below, in the Discussion section to clarify our experimental rationale and discuss the limitations.

Line 346-357:

“Biochemical fractionation strategies (e.g., differential fractionation and density gradient fractionation) combined with antibodies recognizing endogenous proteins are classic in validating the subcellular localization of the protein of interest. Given limited performance of antibodies in detecting SLC35D3, we tried exogenous delivery of SLC35D3 using AAV-PhP.eB, which infected the whole mouse brain efficiently therefore providing satisfactory starting materials. It is worth noting that AAV-PhP.eB potentially results in overexpression of SLC35D3 in the brain that may affect the subcellular distribution of the transporter. In addition, the LP2 fraction (crude SVs) after differential fractionation may contain other organelles such as secretory granules and lysosomes. Subsequent studies using more efficient SLC35D3 antibodies and further purified SVs can be of help to validate the localization of endogenous SLC35D3 in vivo.”

We also toned down the claim throughout the revised manuscript as suggested:

Line 25-26:

Ultrastructural analysis further supported that one of the transporters, SLC35D3, localized to SVs.

Line 105-107:

Further ultrastructural analysis using APEX2-based EM supported that the SLC35D3 is capable of trafficking to SVs.

Line 134-136:

Together, these results indicate that putative vesicular transporters, including a subset of SLC35 family members, likely localize to neuronal SVs.

Line 146-148:

SLC35D3 with relatively low expression level has a higher possibility to localize to synaptic vesicles than to secretory granules.

Line 200-202:

In summary, these data corroborate the view that SLC35D3 is less likely to be a classic ER/Golgi transporter and tends to localize to SVs.

Line 212-213:

…demonstrating that SLC35D3 could localize to SVs in cultured neurons.

Line 309-310:

Here, we report the identification and characterization of three novel SLC35 transporters putatively localized to SVs…

Line 313-315:

These data indicate the potential existence of a novel neuronal vesicular transporter of the nucleotide sugar…

Line 318-323:

We cannot rule out the possibility that these transporters may also play a physiological role in regulated secretory granules in non-neuronal secretory cells. Taking the well-known vesicular transporter VMAT2 as an example, it could localize to both synaptic vesicles and large dense-core vesicles in neurons [6], as well as secretory granules in endocrine cells of the peripheral system [7].

[1] Watson, D. J., Kobinger, G. P., Passini, M. A., Wilson, J. M., and Wolfe, J. H. (2002). Targeted transduction patterns in the mouse brain by lentivirus vectors pseudotyped with VSV, Ebola, Mokola, LCMV, or MuLV envelope proteins. Molecular Therapy, 5(5), 528-537.

[2] Baekelandt, V., Claeys, A., Eggermont, K., Lauwers, E., De Strooper, B., Nuttin, B., and Debyser, Z. (2002). Characterization of lentiviral vector-mediated gene transfer in adult mouse brain. Human gene therapy, 13(7), 841-853.

[3] Baekelandt, V., Eggermont, K., Michiels, M., Nuttin, B., and Debyser, Z. (2003). Optimized lentiviral vector production and purification procedure prevents immune response after transduction of mouse brain. Gene therapy, 10(23), 1933-1940.

[4] Artegiani, B., and Calegari, F. (2013). Lentiviruses allow widespread and conditional manipulation of gene expression in the developing mouse brain. Development, 140(13), 2818-2822.

[5] Chan, K. Y., Jang, M. J., Yoo, B. B., Greenbaum, A., Ravi, N., Wu, W. L., … and Gradinaru, V. (2017). Engineered AAVs for efficient noninvasive gene delivery to the central and peripheral nervous systems. Nature neuroscience, 20(8), 1172-1179.

[6] Nirenberg, M. J., Chan, J., Liu, Y., Edwards, R. H., and Pickel, V. M. (1996). Ultrastructural localization of the vesicular monoamine transporter-2 in midbrain dopaminergic neurons: potential sites for somatodendritic storage and release of dopamine. Journal of Neuroscience, 16(13), 4135-4145.

[7] Weihe, E., Schäfer, M. K. H., Erickson, J. D., and Eiden, L. E. (1994). Localization of vesicular monoamine transporter isoforms (VMAT1 and VMAT2) to endocrine cells and neurons in rat. Journal of Molecular Neuroscience, 5(3), 149-164.

Reviewer #1 (Recommendations for the authors):It would have been helpful to see what happens if the authors had injected the lentivirus construct into the mice and looked for colocalization with markers of synaptic vesicles (VGLUT1, VGAT, synaptophysin as well as performed the fractionation study. It is likely they chose the PHP.eb because of the retro-orbital injection and supposedly good spread of the virus.Otherwise the authors have satisfied my concerns.

We thank the reviewer for raising the concerns. Please refer to our overall responses (Page 1-3) for details.

Reviewer #2 (Recommendations for the authors):Li and colleagues have provided a number of additional experiments and have satisfactorily addressed most of the points. However, one prominent concern remains regarding the subcellular localization of SLC35D3. Essentially, the authors performed a crude synaptosomal preparation and examined a fraction that is enriched in synaptic vesicles and has relatively little ER and Golgi. Nevertheless, the authors overstate the purity of the fraction examined. Indeed, as much as ~30% of SLC35D3 co-localizes with chromogranin A, a canonical secretory granule marker suggesting that this fraction is enriched in secretory granules as well as synaptic vesicles. One strong possibility therefore is that SLC35D3 is not purely found in synaptic vesicles but that it is also localized to other vesicular compartments including secretory granules. The authors did not test enriched the crude synaptic vesicle fractions like LP2 for other compartments like endoscopes, lysosomes etc. raising further questions about the definitive intracellular localization. This would also be in line with the discrepancy concerning the absence of being reported in recent synaptic vesicle proteomics studies such as Taoufiq et al. (2020). Therefore, without obtaining a highly purified synaptic vesicle fraction either through classical biochemical means or even via recently reported immuno-isolation (for example, see Chantranupong et al., eLife 2020), concluding that SLC35D3 is truly a synaptic vesicle transporter cannot be made conclusively.

We thank the reviewer for raising the remaining concerns. Please refer to our overall responses (Page 1-3) for details.

Reviewer #3 (Recommendations for the authors):The authors have responded very thoughtfully to the concerns raised. Questions still persist about the synaptic vesicle localization (still reliant on over-expression and the enrichment by fractionation is not strong), but there is not much more the authors can do given the low expression of endogenous 35D3. Similarly, the lack of bafilomycin effect is surprising since this usually results in run-down of the membrane potential--although the pH gradient may be more resistant (H^+^ efflux requires movement of a counterion). Since other family members differ in mechanism, I suspect this transporter does not depend on a H^+^ electrochemical gradient, accounting for the bafilomycin result, but it is difficult to explain the effect of nigericin as indirect, so again, there is not much more the authors can do and their comparison to another family member here is helpful. Again, it is unclear why the published proteomics has not identified 35D3 but the authors have done all they can and the results presented are reasonable.

We thank the reviewer for the insightful comments.

Reviewer #4 (Recommendations for the authors):The authors have done a thorough job responding to the reviewers' comments.

We are grateful to the reviewer for recommendation.